# The nuclear poly(A)-binding protein Pab2/PABPN1 promotes heterochromatin assembly through the formation of Pab2 nuclear condensates

Ziyue Liu[1], Xiuyi Song[1], Gobi Thillainadesan[2], and Tomoyasu Sugiyama [1]*

**1** School of Life Science and Technology, ShanghaiTech University, Shanghai, China, **2** Biological Sciences Platform, Sunnybrook Research Institute, Toronto, Canada

* tsugiyama@shanghaitech.edu.cn

## Abstract

The assembly of constitutive heterochromatin is a prerequisite for maintaining genome stability. However, the mechanism of heterochromatin formation has yet to be completely understood. Here, we demonstrate a crucial role of the nuclear poly(A)-binding protein (PABP) Pab2/PABPN1 in promoting constitutive heterochromatin formation in the fission yeast *Schizosaccharomyces japonicus*. Histone H3 Lys 9 di- and tri-methylation, hallmarks of heterochromatin, are significantly reduced at centromeres in the absence of Pab2. Pab2 forms nuclear condensates through its RNA-recognition motif (RRM) and the intrinsically disordered domain (IDR), both of which bind to centromeric non-coding RNAs. Intriguingly, two key heterochromatin factors, the histone H3 Lys9 methyltransferase Clr4 and the Mi2-type chromatin remodeler Mit1, associate with centromeres in a Pab2-dependent manner. Pab2 interacts with two putative RNA-binding proteins, the ZC3H3 ortholog Red5 and the RBM26·27 ortholog Rmn1, both essential for heterochromatin formation. Deletion of the Pab2 N-terminal region, which disrupts this interaction, largely abolishes Pab2 function, underscoring the importance of this complex. Pab2 also associates and colocalizes with Ppn1 (a PPP1R10 ortholog), a component of the cleavage and polyadenylation specificity factor (CPSF) complex, and *ppn1* mutations disrupt constitutive heterochromatin. Notably, both Ppn1 and Rmn1 are able to interact with Clr4. Our findings reveal that Pab2 plays a pivotal role in heterochromatin assembly by forming nuclear condensates through its RRM/IDR, and Pab2 condensates facilitate the recruitment of Clr4 and Mit1 to centromeres, potentially through its binding proteins, Ppn1 and Rmn1. This study provides new insights into the mechanisms underlying heterochromatin formation and highlights the importance of RNA-binding proteins and phase separation in this process.

## Author summary

Constitutive heterochromatin is crucial for genome stability, yet its assembly mechanisms are not fully understood. In this study, we investigate the role of the nuclear

**Data availability statement:** RNA-seq data (both raw and processed) can be available at GEO (URL: https://www.ncbi.nlm.nih.gov/geo/). The accession number is GSE237964. Detailed parameters for LC-MS/MS and proteomics data have been deposited to the ProteomeXchange Consortium via the PRIDE partner repository with the dataset identifier PXD053432.

**Funding:** This work was supported by the National Natural Science Foundation of China (General Program: 31971331 to TS) and ShanghaiTech University (Startup fund to TS). The funders had no role in study design, data collection and analysis, decision to publish, or preparation of the manuscript.

**Competing interests:** The authors have declared that no competing interests exist.

poly(A)-binding protein Pab2 (human PABPN1 ortholog) in heterochromatin formation in the fission yeast *Schizosaccharomyces japonicus*. We show that Pab2 is essential for constitutive heterochromatin assembly, as evidenced by reduced H3K9 methylation, a hallmark of heterochromatin, in its absence. Pab2 forms nuclear condensates through its RNA-recognition motif and intrinsically disordered region, both binding to transcripts derived from centromeres; the Pab2 condensates are crucial for Pab2 functions. In addition, the stable association of the H3 Lys 9 methyltransferase Clr4 and the chromatin remodeler Mit1 with centromeres depends on Pab2. In addition, Pab2 interacts with two putative RNA-binding proteins Red5 and Rmn1, essential for heterochromatin formation, and theses interactions are required for its function. Moreover, Pab2 associates with Ppn1, a cleavage and polyadenylation specificity factor (CPSF) complex component, and Ppn1 contributes to heterochromatin integrity. Notably, Ppn1 and Rmn1 interact with the H3 Lys 9 methyltransferase Clr4. Our findings reveal that Pab2 promotes heterochromatin assembly by forming condensates that facilitate Clr4 and Mit1 recruitment to centromeres, potentially through Ppn1 and Rmn1. This highlights the critical role of RNA-binding proteins and phase separation in heterochromatin formation.

## Introduction

Genomic DNA in eukaryotes is organized into two functionally and structurally distinct domains: euchromatin and heterochromatin [1,2]. Euchromatin is characterized by its gene-rich composition, less condensed chromatin structures, and transcriptional activity, whereas heterochromatin is typically gene-poor, more condensed, and largely inactive in transcription and recombination. Heterochromatin can be further classified into two types: constitutive and facultative heterochromatin. Constitutive heterochromatin is found in regions enriched with repetitive DNA sequences, such as centromeres and telomeres. It is stably maintained and plays a crucial role in chromosome segregation and the suppression of recombination and transposons [3–5]. In contrast, facultative heterochromatin is assembled at specific chromosomal loci containing genes that are conditionally expressed in response to developmental and environmental cues. Therefore, the proper assembly of heterochromatin is a prerequisite for maintaining genome integrity and ensuring accurate gene expression.

The fission yeast *Schizosaccharomyces pombe* has been used as a model organism to study the mechanisms of heterochromatin assembly [5–7]. Previous studies have established that RNA interference (RNAi) is the primary pathway for assembling constitutive heterochromatin, and various factors, including DNA-binding proteins and histone deacetylases, are also essential for this process [8–15]. In contrast, the formation of facultative heterochromatin often requires the RNA degradation complex known as MTREC/NURS but does not rely on RNAi [16,17]. Since the mechanisms elucidated in *S. pombe* are conserved across higher eukaryotes [18–20], *S. pombe* has shown its significance in understanding heterochromatin assembly in multicellular organisms.

Despite these advances, the mechanisms underlying heterochromatin formation are not fully understood yet, and further research is necessary to clarify the complex interplay of various factors involved in this process. Notably, another species of fission yeast, *S. japonicus*, has emerged as a potential model for studying heterochromatin [21]. While *S. pombe* features unique repetitive sequences (*dg*, *dh*, and *imr*) at its centromeres, *S. japonicus* contains Gypsy-type retrotransposable elements that are comparable to those found in human centromeric structures [22]. Moreover, RNAi is essential for both *S. japonicus* and mammals [23,24] but not for *S. pombe* [12], suggesting that RNAi-dependent heterochromatin assembly

at centromeres is indispensable for cell viability in *S. japonicus* and mammals. Based on these findings, *S. japonicus* appears to have its unique strength in the study of heterochromatin.

Recent evidence has underscored the importance of polyadenylation in RNAi-mediated heterochromatin assembly. For example, Rdp1, an RNA-dependent RNA polymerase essential for RNAi, interacts with the non-canonical poly(A) polymerase Cid12, and both Rdp1 and Cid12 are indispensable for centromeric heterochromatin [12,13,25]. In addition, CPSF (cleavage and polyadenylation specificity factor), which plays a role in mRNA polyadenylation, has been implicated in building centromeric heterochromatin [26,27]. Moreover, the *S. pombe* Caprin protein Cpn1, which is required for efficient heterochromatin establishment, is functionally linked to the cytoplasmic poly(A)-binding protein Pabp [28]. Besides that, Rde-3 — a member of the polymerase beta nucleotidyltransferase superfamily — is necessary for RNAi in *C. elegans* [29], and Dicer-2 associates with the cytoplasmic poly(A) polymerase Wispy in fruit flies [30]. These findings suggest that the functional connection between polyadenylation and RNAi is evolutionarily conserved. Nonetheless, the role of polyadenylation in heterochromatin assembly has not been extensively studied.

In this study, we present our findings on the nuclear poly(A)-binding protein Pab2/PABPN1 in *S. japonicus*. Our analyses demonstrated that Pab2 promotes selective mRNA degradation in vegetative *S. japonicus*. Intriguingly, we observed an accumulation of centromeric transcripts in *pab2* deletion cells (*pab2Δ*), which was further supported by a reduction in histone H3 Lys9 methylation — the hallmark of heterochromatin — in *pab2Δ* cells. These observations indicate that Pab2 has a crucial role in assembling constitutive heterochromatin in *S. japonicus*, particularly intriguing since Pab2 is dispensable for constitutive heterochromatin formation in *S. pombe* [31]. Furthermore, we identified two RNA-binding proteins — Red5/ZC3H3 and Rmn1/RBM26·27 — as well as Ppn1/PPP1R10, a subunit of CPSF, all of which interacts with Pab2 and are essential for centromeric heterochromatin assembly. Importantly, Pab2 facilitates the recruitment of two key proteins, the histone H3 Lys9 methyltransferase Clr4 and the Mi2-type chromatin remodeler Mit1, to centromeres. Our analyses also highlight the importance of the Pab2 C-terminal intrinsically disordered region and the Pab2 RNA recognition motif (RRM), both vital for Pab2 nuclear condensates formed via phase separation, in promoting heterochromatin assembly. Based on these results, we propose that Pab2 creates a nuclear microenvironment that recruits key heterochromatin factors to centromeric regions, thereby facilitating the assembly of constitutive heterochromatin.

## Results

### Pab2 is necessary for proper growth and meiosis

Given the importance of polyadenylation and poly(A)-binding proteins in various cellular transactions, we first constructed a *pab2* (*SJAG_04601*) deletion strain (*pab2Δ*), in the fission yeast *Schizosaccharomyces japonicus* and characterized the *pab2Δ* cells. The *pab2Δ* cells exhibited slow growth, particularly pronounced at lower temperatures (18°C and 26°C) compared to wild-type (WT) cells (S1A and S1B Fig). This observation suggests that *S. japonicus* Pab2 has a critical role in growth, which may not be evident in *S. pombe*, where *pab2* deletion does not significantly hinder growth at 30°C [32]. In addition, microscopic analyses demonstrated that approximately 10% of *pab2Δ* cells displayed an abnormal DAPI staining pattern (S1C Fig), suggesting that Pab2 is required for normal mitotic cell cycle progression. Moreover, the number of asci produced by homothallic *pab2Δ* cells was reduced compared to that of homothallic WT cells when nitrogen sources were depleted (S1D Fig), and the mating efficiency was significantly decreased in *pab2Δ* cells (S1E Fig). These results indicate that Pab2 is required for efficient mating in *S. japonicus*, while Pab2 is dispensable for efficient mating in *S. pombe*

[33]. Our findings demonstrate that Pab2 is essential for proper growth and meiosis in *S. japonicus*, revealing a novel role for this poly(A)-binding protein in these fundamental cellular processes and highlighting a potential divergence between the two fission yeast species.

## The nuclear poly(A)-binding protein Pab2 prevents the untimely accumulation of putative meiotic genes in *S. japonicus*

Pab2 and its orthologs play a role in RNA metabolism [16,34–36]. To investigate the function(s) of Pab2 in mitosis and meiosis, transcriptome analysis of *pab2Δ* was performed using two rounds of RNA sequencing. The RNA-seq results were reproducible, with highly significant overlaps showing more than a 2-fold increase and less than a 0.5-fold decrease in *pab2Δ* between two independent experiments (S1F Fig). We identified a total of 389 and 83 transcripts that were upregulated and downregulated in *pab2Δ*, respectively, across both biological replicates (S1 and S2 Tables). Gene ontology (GO) analysis revealed that 22 out of 389 of the Pab2-suppressed genes are meiosis-related ($p < 10^{-3.28}$) (Fig 1A and S3 Table), indicating that Pab2 plays a role in repressing potential meiosis-related gene expression. Consistent with previous findings in *S. pombe* showing that Pab2 suppresses untimely expression of meiotic genes [33,36], RT-qPCR analysis confirmed elevated mRNA levels for four examined putative meiotic genes in *pab2Δ* (Fig 1B). In contrast, GO analysis of the downregulated genes suggested that Pab2 promotes the expression of genes involved in small molecule (amino acids) metabolic/catabolic processes (S4 Table), which likely contributes to normal vegetative growth of *S. japonicus* (S1A and S1B Fig).

We previously reported that the Zn-finger protein Red1 and its associated proteins promote selective degradation of meiotic mRNAs during vegetative growth in *S. pombe* [16,37–39]. As anticipated, the levels of three putative meiotic mRNAs (*mei4*, *rec8*, and *spo5*) were accumulated in *S. japonicus red1Δ* cells (S1G Fig). Moreover, Pab2 colocalizes with Red1 in nuclear foci (S1H Fig), as reported in *S. pombe* [39,40], suggesting that Pab2 and Red1 nuclear foci are important primarily for putative meiotic mRNA degradation. These data indicate a conserved role for Pab2 and Red1 in selective mRNA degradation in both *S. japonicus* and *S. pombe*.

## Pab2 participates in the suppression of transcripts derived from constitutive heterochromatin

We identified that transcripts from centromeric and telomeric domains were upregulated in *pab2Δ* (Fig 2A and S1 Table). Transcripts with elevated levels included retrotransposon fragments within constitutive heterochromatin and three genes embedded within the mating-type locus (*SJAG_05153*, *SJAG_05154*, and *SJAG_01113*). This finding was unexpected as, to the best of our knowledge, Pab2 has not previously been directly implicated in the formation of constative heterochromatin in either *S. japonicus* or *S. pombe*. Consistent with our RNA-seq data, RT-qPCR confirmed that the transcript levels from centromeric and telomeric regions (S2A Fig) were significantly elevated in *pab2Δ* (Fig 2B). Because transcripts from repetitive sequences are critical for RNA-based heterochromatin assembly in fission yeast [24,41,42], we hypothesized that heterochromatin assembly was adversely affected in *S. japonicus* lacking Pab2.

To investigate heterochromatin in *S. japonicus*, we first isolated *clr4–1*, a temperature-sensitive mutant of the histone H3 Lys9 methyltransferase Clr4/SUV39 (S2B Fig), as a control strain since Clr4 is essential for *S. japonicus* viability [24]. Genome sequencing results indicated that the *clr4–1* mutant carried two nucleotide substitutions: one leading to an amino acid substitution (TGT to CGT, Cys344Arg) in the SET domain responsible for histone H3

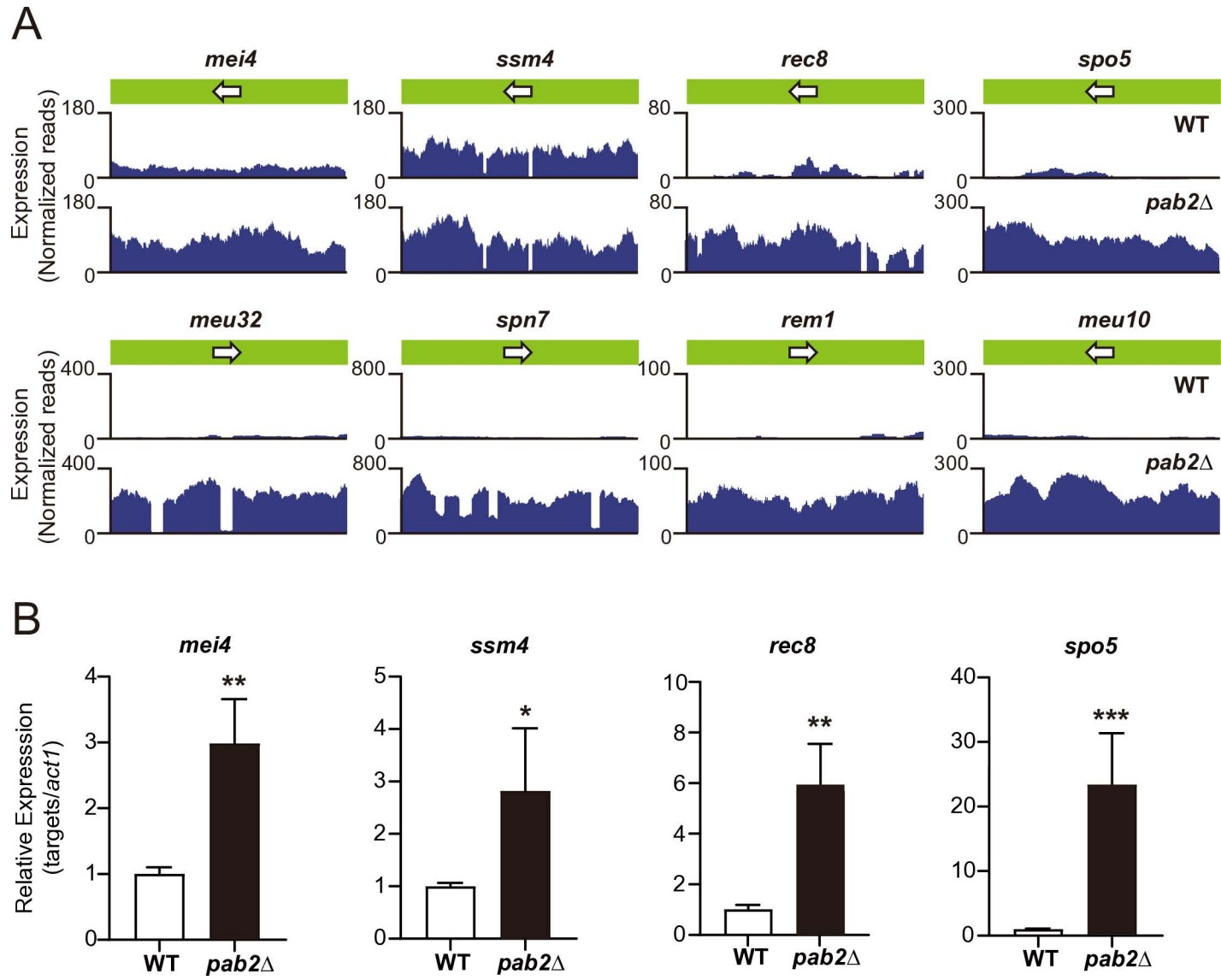

**Fig 1. Putative meiotic mRNA accumulation in *S. japonicus pab2Δ*.** (A) Expression of putative meiosis-related genes in wild-type (WT) and the *pab2Δ* cells. RNA-seq data of four representative meiotic genes (top) and four other highly expressed meiotic genes (bottom) are shown. (B) RT-qPCR data from four putative meiotic genes in WT and *pab2Δ* cells. The four putative meiotic mRNAs (*mei4*, *ssm4*, *rec8*, and *spo5*) were normalized to *act1* mRNA to determine their relative expression levels. Data represent mean ± SD from three independent experiments. Statistical significance was determined using a two-tailed unpaired *t*-test (*$p < 0.05$, **$p < 0.01$, and ***$p < 0.005$).

Lys9 methyltransferase activity, and the other being a silent mutation (ACA to ACG, Thr51) in the chromodomain (S2C Fig). Additionally, RNA-seq data demonstrated that the levels of both centromeric (*SJATN_00017*) and telomeric (*SJAG_03765* and *SJAG_03766*) transcripts increased substantially in *clr4–1* (Fig 2A and S5 Table), and the increase was confirmed by RT-qPCR (Fig 2B). GO analysis indicated that transcription levels of genes involved in the amino acid metabolic process were reduced in *clr4–1* (S6 Tables).

Interestingly, mRNAs derived from three genes encoding Mc-like (*SJAG_05153*), Pc-like (*SJAG_01113*), and Pi (*SJAG_05154*) proteins within the mating-type locus were not accumulated in *clr4–1* (Fig 2A), despite their location within pericentromeric heterochromatin of

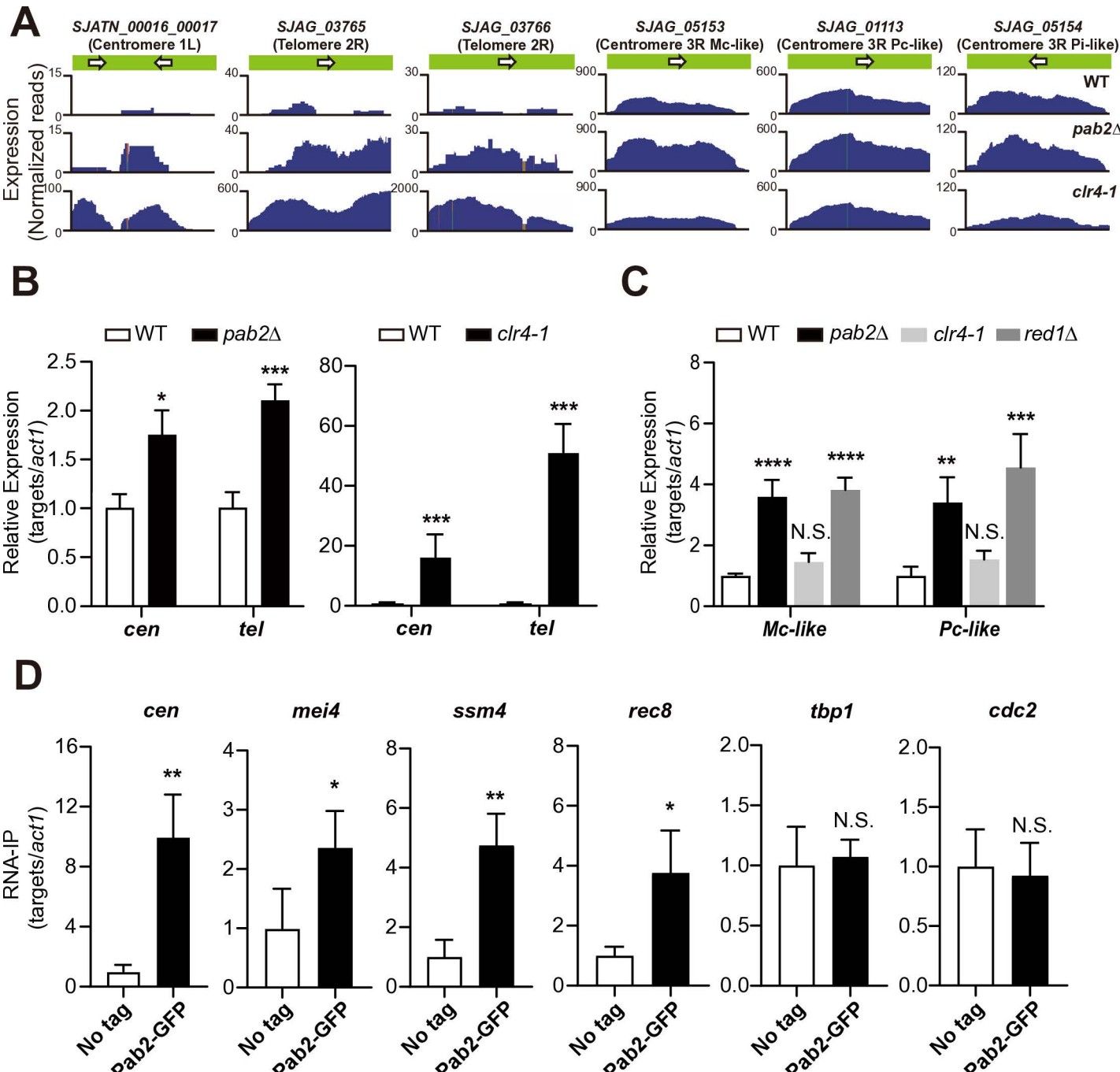

**Fig 2. Pab2 contributes to suppressing transcripts derived from constitutive heterochromatin.** (A) RNA-seq read coverage of transcripts derived from centromeres, telomeres, and the mating-type locus (located within the pericentromeric heterochromatin of chromosome III) in wild-type (WT, top panels), *pab2Δ* (middle panels), and *clr4–1* (bottom panels) strains. Note that the *clr4–1* panels use a different scale for normalized reads (RPM). (B) RT-qPCR data from centromeric and telomeric transcripts in WT, *pab2Δ*, and *clr4–1* strains. Centromeric (*cen*) and telomeric transcripts (*tel*) were normalized to *act1* mRNA to determine their relative expression levels. Data represent mean ± SD from three independent experiments. Statistical significance was determined using a two-tailed unpaired *t*-test (*$p < 0.05$ and ***$p < 0.005$). (C) RT-qPCR data from mating-type transcripts in WT, *pab2Δ*, *clr4–1*, and *red1Δ* strains. Transcripts from Mc-like and Pc-like genes (*Mc-like* and *Pc-like*) were normalized to *act1* mRNA to determine their relative expression levels. Data represent mean ± SD from three independent experiments. Statistical significance was determined using a one-way ANOVA followed by Dunnett's multiple comparison test, with WT as the reference sample (**$p < 0.01$, ***$p < 0.001$, and ****$p < 0.0001$; N.S.: not significant). (D) RNA immunoprecipitation (RNA-IP) using anti-GFP antibody in Pab2-GFP and untagged parental (No tag) strains. Precipitated RNAs were subjected to RT-qPCR. Centromeric RNA (*cen*) and five other mRNAs (*mei4*, *ssm4*, *rec8*, *tbp1*, and *cdc2*) were normalized to *act1* mRNA to determine their relative enrichment. Data represent mean ± SD from three independent experiments. Statistical significance was determined using a two-tailed unpaired *t*-test (*$p < 0.05$ and **$p < 0.01$; N.S.: not significant).

chromosome III [21]. We hypothesized that these mRNAs are degraded by the Pab2/Red1-dependent RNA elimination pathway due to their association with meiosis. RT-qPCR results revealed that mRNAs encoding Mc-like and Pc-like proteins were upregulated in both *pab2Δ* and *red1Δ*, but not in *clr4–1* (Fig 2C), indicating that the genes within the mating-type locus are primarily repressed by the RNA elimination mechanism in *S. japonicus*.

Our RNA-seq data also demonstrated that Pab2- and Clr4-target genes significantly overlapped (S2D Fig and S7 and S8 Tables), suggesting that Pab2 and Clr4 cooperated to suppress and promote the expression of specific genes. GO analysis of the downregulated genes in both *pab2Δ* and *clr4–1* mutant strains showed enrichment in gamma-aminobutyric acid metabolic process, small molecule catabolic process, gamma-aminobutyric acid catabolic process, non-proteinogenic amino acid metabolic process, and arginine biosynthetic process via ornithine (S2E Fig). On the other hand, no significant enrichment was found in GO analysis of the upregulated genes in both *pab2Δ* and *clr4–1* mutant strains.

A previous study demonstrated that non-coding RNAs derived from centromeres are transcribed by RNA polymerase II (Pol II) in *S. pombe* [43], prompted us to examine Pol II occupancy at centromeres in *pab2Δ*. ChIP-qPCR analysis revealed no significant increase in Pol II occupancy in *pab2Δ* (S2F Fig). This is consistent with the modest upregulation of heterochromatic transcripts observed in *pab2Δ* (Fig 2A and 2B), particularly when compared to the substantial increase in both Pol II occupancy (S2F Fig) and centromeric transcript levels (Fig 2A and 2B) observed in *clr4–1*. These findings suggest that transcriptional gene silencing at heterochromatin is largely maintained even in *pab2Δ*.

Given that Pab2 is an RNA-binding protein, we assumed that it binds to transcripts derived from constitutive heterochromatin and putative meiotic genes. To examine this assumption, we performed RNA immunoprecipitation (RNA-IP) followed by RT-qPCR. As shown in Fig 2D, Pab2 preferentially bound to RNAs derived from centromeres and putative meiotic mRNAs but did not preferentially associate with *tbp1* and *cdc2* mRNAs, which encode the TATA-binding protein Tbp1/TBP and the cyclin-dependent kinase Cdc2/Cdk1, respectively. These findings suggest that Pab2 selectively interacts with certain RNAs.

## Pab2 facilitates H3K9 methylation at constitutive heterochromatin

One of the hallmarks of heterochromatin is histone H3 Lys9 di- and tri-methylation (H3K9me2/H3K9me3), which provides binding sites for chromodomain proteins such as HP1 [4,5]. While both H3K9me2 and H3K9me3 are repressive marks that are bound by chromodomain proteins, H3K9me3 is generally associated with a stronger and more stable form of transcriptional repression, often leading to heterochromatin formation, compared to H3K9me2 [44]. Since heterochromatic silencing is partially disrupted in *pab2Δ*, we conducted chromatin immunoprecipitation followed by qPCR (ChIP-qPCR) to assess H3K9me levels in *pab2Δ*. ChIP-qPCR results revealed that H3K9me2 at centromeres and telomeres was significantly decreased in *pab2Δ* and *clr4–1* (Fig 3A). In contrast, H3K9me3 was significantly reduced at centromeres but not at telomeres in *pab2Δ* and *clr4–1* (Fig 3A), suggesting the more robust or redundant mechanism of heterochromatin assembly at telomeres. These results suggest that Pab2 is not absolutely required for H3K9me, but it promotes proper H3K9me in *S. japonicus*.

As described above, *S. pombe* has facultative heterochromatin at various loci, with two representative heterochromatin islands found at two meiotic gene loci, *mei4+* and *ssm4+* [7,45–47]. Because the genomic regions surrounding these loci are syntenic between *S. japonicus* and *S. pombe*, we reasoned that *S. japonicus* might also have heterochromatin islands at the orthologous loci. However, we did not observe any significant enrichment of H3K9me2 at either locus by ChIP-qPCR (Fig 3B), suggesting that the assembly of facultative

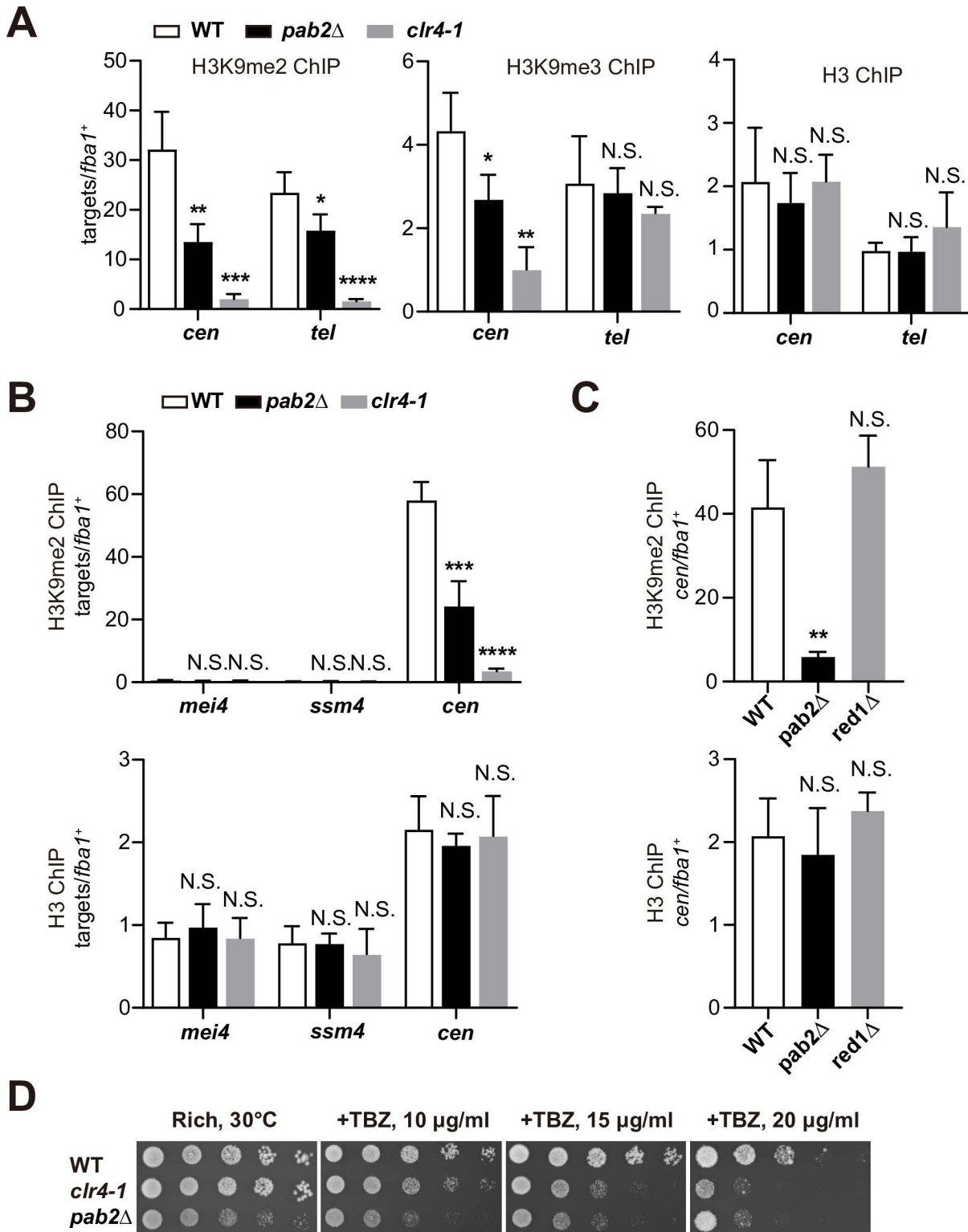

**Fig 3. Pab2 plays an important role in constitutive heterochromatin.** (A) H3K9me2, H3K9me3, and total H3 enrichment at centromeres and telomeres in wild-type (WT), *pab2Δ*, and *clr4–1* cells were assessed by ChIP-qPCR. Data represent mean ± SD from three independent experiments. Statistical significance was determined using a one-way ANOVA followed by Dunnett's multiple comparison test, with WT as the reference sample (*$p < 0.05$, **$p < 0.01$, ***$p < 0.001$, and ****$p < 0.0001$; N.S.: not significant). (B) No noticeable heterochromatin islands at the *mei4+* and *ssm4+* loci in *S. japonicus*. ChIP-qPCR analysis of H3K9me2 and total H3 enrichment at these loci and centromeres in WT,

*pab2Δ*, and *clr4–1* was conducted. Statistical significance was determined using a one-way ANOVA followed by Dunnett's multiple comparison test, with WT as the reference sample (\*\*\*$p < 0.001$ and \*\*\*\*$p < 0.0001$; N.S.: not significant). (C) ChIP-qPCR analysis of H3K9me2 and total H3 enrichment at centromeres in WT, *pab2Δ*, and *red1Δ*. Data represent mean ± SD from three independent experiments. Statistical significance was determined using a one-way ANOVA followed by Dunnett's multiple comparison test, with WT as the reference sample (\*\*$p < 0.01$; N.S.: not significant). (D) Thiabendazole (TBZ) sensitivity assay of WT, *pab2Δ*, and *clr4–1* cells. Ten-fold serial dilutions were spotted onto complete medium plates in the presence or absence of TBZ and grown at 30°C.

heterochromatin at meiotic genes is not conserved even among *Schizosaccharomyces* species. Note that facultative heterochromatin at the meiotic genes is not always observed even in WT cells of *S. pombe* [48]. Besides, considering that both Pab2 and Red1 suppressed inferred meiotic mRNAs during vegetative growth (Figs 1 and S1G), we propose that Pab2, but not Red1, has two separable functions: mRNA degradation and heterochromatin assembly.

Previous studies reported that RNA degradation proteins, including Red1, Mmi1, and Erh1, were crucial for the assembly of facultative but not constitutive heterochromatin in *S. pombe* [17,31,37,45,47,49]. Similarly, the levels of H3K9me2 at centromeres in *red1Δ* were comparable to those in WT (Fig 3C), indicating that the Red1-dependent RNA degradation pathway is dispensable for the formation of constitutive heterochromatin in *S. japonicus*.

The expected increase in centromeric transcripts was not observed in *pab2Δ* (Fig 2B), although H3K9me2/H3K9me3 were significantly decreased (Fig 3A). We hypothesized that Pab2-independent pathways might degrade the centromeric transcripts. Indeed, centromeric transcripts were also elevated in *cid14Δ*, *S. japonicus* lacking Cid14/TENT4 — a constituent of the nuclear exosome-targeting complex TRAMP [50], but the increase was more pronounced in *cid14Δ* (S3A Fig). This result indicates that TRAMP-dependent RNA decay suppresses centromeric transcripts at the post-transcription level, as described in *S. pombe* [50]. Additionally, combining the *pab2*<sup>RRM3A</sup> mutation (see below) with *cid14Δ* resulted in a significant reduction of H3K9me2 (S3B Fig). Thus, it is plausible that centromeric transcripts upregulated in *pab2Δ* are degraded by the TRAMP-mediated RNA elimination system, thereby masking the impact of *pab2* deletion.

It has been demonstrated that centromeric heterochromatin is vital for proper chromosome segregation, and loss of heterochromatin resulted in sensitivity to thiabendazole (TBZ), a microtubule-destabilizing agent [51–53]. If the partial loss of H3K9me2/H3K9me3 is biologically significant, we would expect TBZ sensitivity to be observed in *pab2Δ* cells. Indeed, we found that both *pab2Δ* and *clr4–1* exhibited sensitivity to TBZ (Fig 3D). From these results, we proposed that Pab2 is essential for the assembly of constitutive heterochromatin in *S. japonicus*.

## The Pab2 intrinsically disordered region, which coordinates phase separation, is required for heterochromatin assembly

Intrinsically disordered regions (IDRs) are essential for forming biomolecular condensates via phase separation [54,55]. During vegetative growth in *S. japonicus*, Pab2 predominantly forms a single nuclear focus (S1H and S4A Figs), consistent with observations in *S. pombe* [33,40], suggesting that Pab2 possesses an IDR that drives condensate formation. Indeed, DISOPRED3 [56] predicted that the C-terminal domain of Pab2 (amino acids: 136–174), rich in arginine and glycine residues, is an intrinsically disordered region (Fig 4A). Treatment with 1,6-hexanediol (1,6-HD), which disrupted membrane-less biological condensates [57,58], abolished Pab2 foci, and these foci reappeared upon 1,6-HD removal (S4B Fig). As 1,6-HD treatment did not affect steady-state Pab2-GFP levels (S4C Fig), these results indicate that Pab2 forms phase-separated nuclear condensates in a C-terminal IDR-dependent manner.

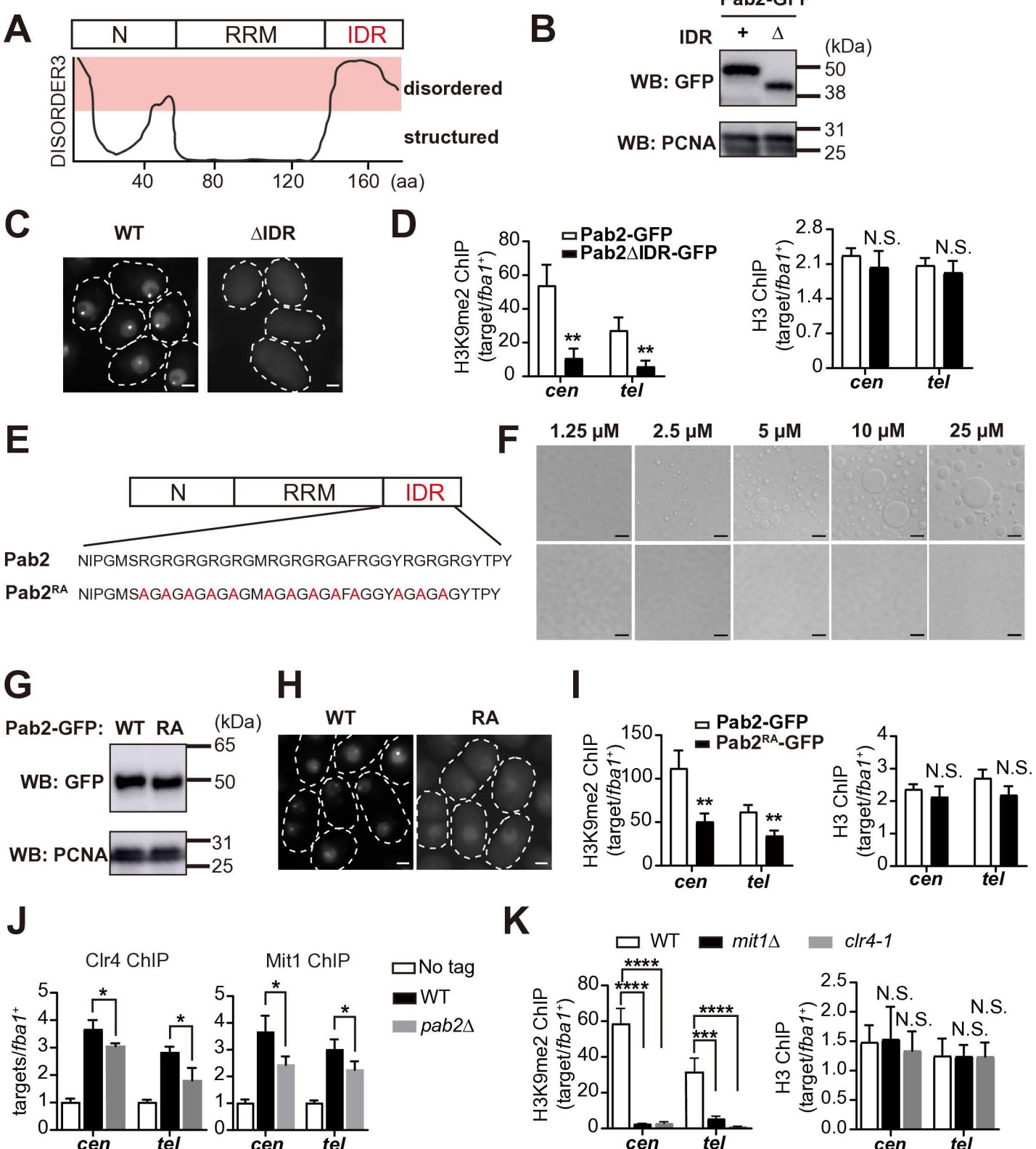

**Fig 4. The Pab2 intrinsically disordered region (IDR) is required for heterochromatin assembly and condensate formation.** (A) Prediction of Pab2 IDR using DISOPRED3. The C-terminal region of Pab2 (amino acids: 136–174) is predicted to be intrinsically disordered. The N-terminal domain (N) is predicted to be partially disordered, while the RNA-recognition motif (RRM) is predicted to be ordered. (B) Western blot analysis of Pab2-GFP and Pab2ΔIDR-GFP protein levels. Pcn1/PCNA serves as a loading control. (C) Fluorescent microscopy of Pab2-GFP (WT) and Pab2ΔIDR-GFP (ΔIDR) localization. The white dotted lines indicate

cell boundaries. Scale bars, 2 µm. (D) ChIP-qPCR analysis of H3K9me2 and total H3 enrichment at centromeres and telomeres. Data represent mean ± SD from three independent experiments. Statistical significance was determined using a two-tailed unpaired $t$-test (**$p < 0.01$; N.S.: not significant). (E) Schematic representation and amino acid sequences of the IDR of Pab2 and Pab2RA mutant proteins. (F) *in vitro* droplet formation assay using purified Pab2 and Pab2RA proteins. Scale bars, 5 µm. (G) Western blot analysis of Pab2-GFP (WT) and Pab2RA-GFP (RA). Pcn1/PCNA serves as a loading control. (H) Fluorescent microscopy of Pab2-GFP (WT) and Pab2RA-GFP (RA) localization. The white dotted lines indicate cell boundaries. Scale bars, 2 µm. (I) ChIP-qPCR analyses of H3K9me2 and total H3 enrichment at centromeres and telomeres. Data represent mean ± SD from three independent experiments. Statistical significance was determined using a two-tailed unpaired $t$-test (**$p < 0.01$; N.S.: not significant). (J) ChIP-qPCR analyses of 3 × FLAG-Clr4 and Mit1-GFP levels at centromeres and telomeres in WT and *pab2Δ* strains. An untagged strain (No tag) was used as a negative control. Data represent mean ± SD from three independent experiments. Statistical significance was determined using a two-tailed unpaired $t$-test (*$p < 0.05$). (K) ChIP-qPCR analysis of H3K9me2 and total H3 enrichment at centromeres and telomeres in WT, *mit1Δ*, and *clr4–1*. Data represent mean ± SD from three independent experiments. Statistical significance was determined using a one-way ANOVA followed by Dunnett's multiple comparison test, with WT as the reference sample (***$p < 0.001$ and ****$p < 0.0001$; N.S.: not significant).

Given the roles of phase separation in chromatin dynamics [59–61], we investigated whether Pab2 condensates contribute to heterochromatin assembly. To this end, we generated a strain expressing Pab2ΔIDR-GFP, a GFP-tagged Pab2 protein lacking the C-terminal IDR (S4D Fig). Western blotting confirmed comparable expression levels of Pab2-GFP and Pab2ΔIDR-GFP (Fig 4B). However, microscopic analysis revealed that Pab2ΔIDR-GFP failed to either localize to the nucleus or form nuclear foci (Fig 4C), indicating that the IDR is essential for proper Pab2 localization. ChIP-qPCR demonstrated reduced H3K9me2 levels at centromeres and telomeres in the Pab2ΔIDR-GFP strain (Fig 4D). Consistent with the partial disruption of constitutive heterochromatin, Pab2ΔIDR-GFP cells were more sensitive to TBZ than WT cells (S4E Fig). Moreover, we observed significantly increased levels of putative meiotic mRNAs, but not centromeric or telomeric transcripts, in Pab2ΔIDR-GFP cells (S4F Fig). These observations suggest that the Pab2 C-terminal domain is crucial for both selective mRNA decay and heterochromatin assembly. However, the data also suggest that TRAMP-dependent RNA decay pathway can compensate for the loss of Pab2 in degrading heterochromatic transcripts.

Arginine (Arg) residues facilitate interactions among disordered domains via charge-charge interactions and cation-π interactions [62], and these interactions stabilize protein/RNA condensates through the multivalency of the guanidinium group [63]. We therefore investigated whether Arg residues within the Pab2 IDR (Fig 4E) are crucial for condensate formation. We expressed and purified the MBP (maltose-binding protein)-Pab2–6 × His recombinant protein in *E. coli* (S4G and S4H Fig). Under our experimental conditions, recombinant Pab2 formed spherical structures at 2.5 µM (Fig 4F) that could fuse with each other (S1 Movie), demonstrating its ability to form liquid-like droplets *in vitro*. We also purified recombinant Pab2RA, in which all Arg residues in the IDR were replaced with Ala (Fig 4E). Recombinant Pab2RA protein formed spherical structures only at significantly higher concentration (25 µM) compared to wild-type Pab2 (Fig 4F), indicating that the Arg residues are critical for efficient condensate formation *in vitro*.

To assess the role of these Arg residues in vivo, we constructed a strain expressing Pab2RA-GFP. This mutation did not adversely affect steady-state Pab2-GFP protein levels (Fig 4G). However, Pab2RA-GFP failed to form nuclear foci, exhibiting diffuse nuclear localization instead (Fig 4H). This confirms that Arg residues in the IDR are indispensable for Pab2 condensate assembly *in vivo*. In addition, Pab2RA-GFP cells exhibited significantly decreased H3K9me2 levels at centromeres and telomeres (Fig 4I) and elevated sensitivity to TBZ than Pab2-GFP cells (S4E Fig). Moreover, putative meiotic mRNA levels increased, while heterochromatin transcripts remained unchanged in Pab2RA-GFP cells (S4I Fig), mirroring the observations in Pab2ΔIDR-GFP cells. These results underscore the crucial role of the IDR in heterochromatin assembly.

Given that biomolecular condensates can increase the local concentration of specific factors [64–66], we examined whether Pab2 condensates recruit or stabilize the association of heterochromatin factors, such as Clr4, at heterochromatin. ChIP-qPCR analysis revealed significantly reduced Clr4 binding to constitutive heterochromatin in *pab2Δ* (Fig 4J). We also observed decreased levels of Mit1 (SJAG_03982), a Mi-2-like chromatin remodeler in SHREC [15,67], at centromeres and telomeres in *pab2Δ* (Fig 4J). The importance of Mit1 in heterochromatin assembly was supported by the substantial loss of H3K9me2 and increased TBZ sensitivity in *mit1Δ* cells (Figs 4K and S4J). These findings suggest that Pab2 promotes heterochromatin assembly by facilitating the recruitment of Clr4 and Mit1 to constitutive heterochromatin through its condensates.

## Pab2 RNA-binding activity is critical for condensate formation and heterochromatin assembly

While we observed decreased H3K9me enrichment at centromeres and telomeres of *pab2Δ* (Fig 3A), it remained unclear whether the RNA-binding activity of Pab2 is necessary for heterochromatin assembly. To investigate this, we constructed three mutant strains: Pab2ΔRRM, expressing Pab2 lacking the RNA recognition motif (RRM); Pab2$^{RRM3A}$, carrying the substitutions of three aromatic residues (Y61A, F101A, and Y103A) critical for RRM activity [68]; and Pab2$^{RRM3A-RA}$, possessing both the RRM mutations and multiple Arg to Ala substitutions in the IDR (Fig 5A). Western blot analysis demonstrated that the steady-state levels of Pab2ΔRRM, Pab2$^{RRM3A}$, and Pab2$^{RRM3A-RA}$ were comparable to that of Pab2 (Fig 5B). However, we did not readily observe nuclear foci in cells expressing any of these mutant proteins (Fig 5C), indicating that the formation of Pab2 foci depends on RRM. To examine whether each Pab2 mutant protein binds to centromeric transcripts, we performed RNA-IP. As shown in Fig 5D, Pab2 and Pab2$^{RA}$ preferentially bound to centromeric transcripts, whereas Pab2ΔRRM, Pab2$^{RRM3A}$, and Pab2$^{RRM3A-RA}$ lost their ability to associate with centromeric RNAs. These observations highlight the importance of the RRM for Pab2 RNA-binding and for Pab2 nuclear condensate formation.

We then conducted RT-qPCR to assess RNA expression in the Pab2 mutant strains. We found increased transcript levels in the following strains: *pab2ΔRRM* showed elevated levels of transcripts from *mei4$^+$*, *ssm4$^+$*, *spo5$^+$*, *rec8$^+$*, and centromeres; *pab2$^{RRM3A}$* exhibited increases in transcripts derived from *ssm4$^+$* and *rec8$^+$*; and in *pab2$^{RRM3A-RA}$* had elevated levels of transcripts from *mei4$^+$*, *ssm4$^+$*, *spo5$^+$*, *rec8$^+$*, and centromeres (S5A Fig). Additionally, H3K9me2 ChIP-qPCR revealed reduced levels of H3K9me2 at centromeres in all the Pab2 mutant strains, as well as in *pab2Δ* cells (Fig 5E). Consistent with this observation, all Pab2 mutant strains exhibited increased sensitivity to TBZ compared to WT cells (S5B Fig), indicating that the loss of heterochromatin in these mutants is biologically significant. Thus, we conclude that the RRM of Pab2 is critical for its functions in heterochromatin assembly and silencing of putative meiotic genes.

## Pab2 preferentially associates with centromeres in both RRM and IDR-dependent manners

Pab2 promotes H3K9 methylation at constitutive heterochromatin and forms nuclear condensates, suggesting that Pab2 directly associates with or is localized in proximity to heterochromatin. To initially investigate this potential association, we examined the colocalization of Pab2 with known heterochromatin markers. Fluorescent microscopy demonstrated the minimal colocalization of Pab2 with Chp1 (S5C Fig), a chromodomain protein that binds to H3K9me [69]. Considering this result with the earlier observation that Pab2 foci colocalize

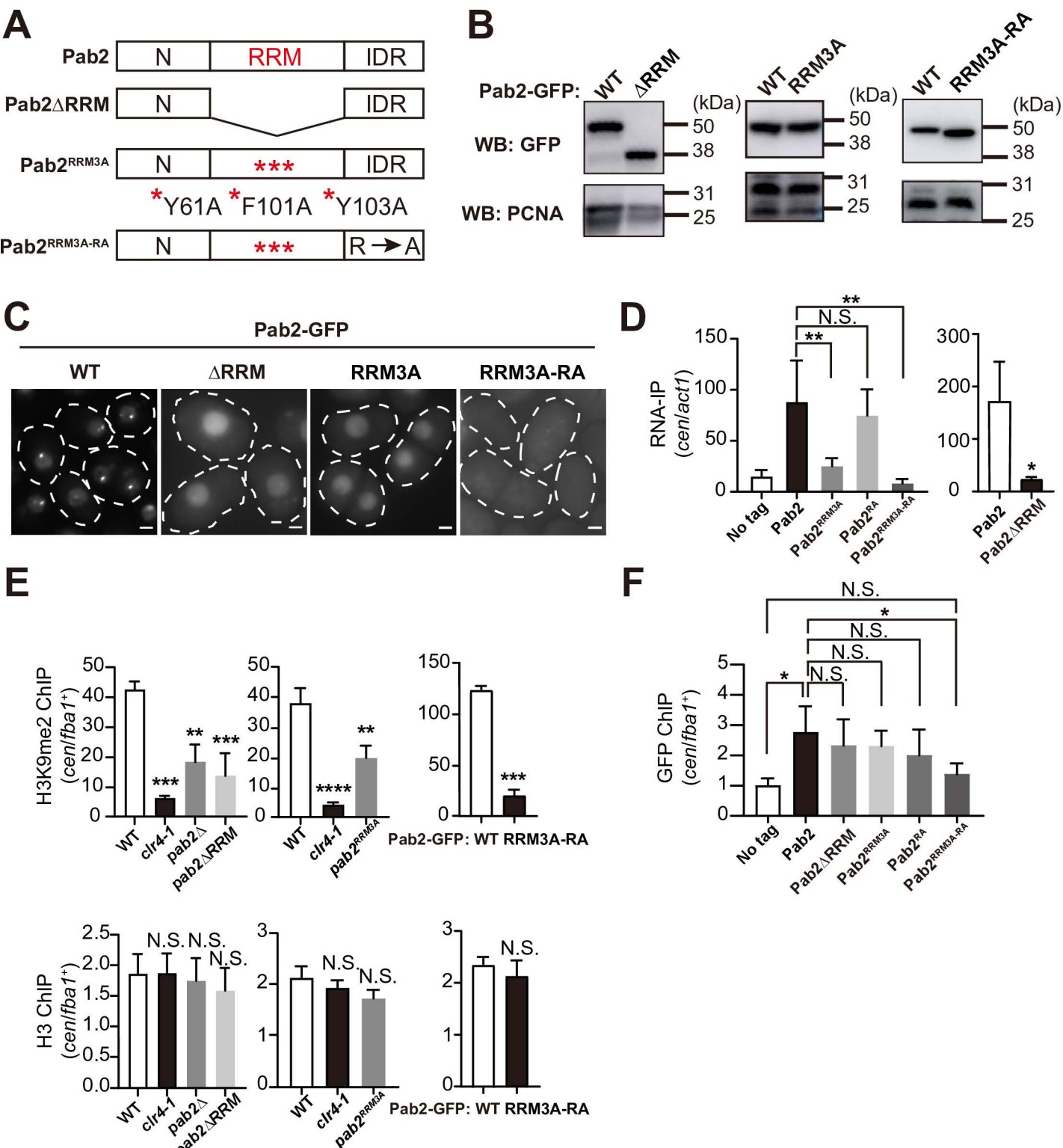

**Fig 5. The Pab2 RNA-recognition motif (RRM) is required for heterochromatin assembly.** (A) Schematic representation of Pab2 constructs: Pab2ΔRRM lacks the RRM; Pab2RRM3A contains three amino acids substitutions (Y61A, F101A, and Y103A); Pab2RRM3A-RA combines the RRM mutations of Pab2RRM3A with additional arginine-to-alanine mutations in the IDR. (B) Western blot analysis of Pab2-GFP (WT), Pab2ΔRRM-GFP (ΔRRM), Pab2RRM3A-GFP (RRM3A), and Pab2RRM3A-RA-GFP (RRM3A-RA) with Pcn1/PCNA as a loading control. (C) Fluorescence microscopy images of Pab2-GFP and its RRM mutants. The white dotted lines indicate cell shapes. Scale bars, 2 μm. (D) RNA immunoprecipitation (RNA-IP) followed by RT-qPCR to assess RNA-binding activity in cells expressing untagged Pab2 (No tag),

Pab2-GFP, Pab2ΔRRM-GFP, Pab2[RRM3A]-GFP, Pab2[RA]-GFP, or Pab2[RRM3A-RA]-GFP. Centromeric RNA (*cen*) was normalized to *act1* mRNA to determine their relative enrichment. Data are presented as mean ± SD from three independent trials. (Left) Statistical significance was determined using a one-way ANOVA followed by Dunnett's multiple comparison test, with 'Pab2' as the reference sample (**$p < 0.01$; N.S.: not significant). (Right) Statistical significance was determined using a two-tailed unpaired *t*-test (*$p < 0.05$). (E) ChIP-qPCR analysis of H3K9me2 and total H3 enrichment at centromeres using the indicated strains. Data are presented as mean ± SD from three independent experiments. (Left and middle) Statistical significance was determined using a one-way ANOVA followed by Dunnett's multiple comparison test, with WT as the reference sample (**$p < 0.01$, ***$p < 0.001$ and ****$p < 0.0001$; N.S.: not significant). (Right) Statistical significance was determined using a two-tailed unpaired *t*-test (***$p < 0.005$; N.S.: not significant). (F) ChIP-qPCR analysis of Pab2 and i*t*s mutant proteins at centromeres. Data are presented as mean ± SD from three independent experiments. Statistical significance was determined using a one-way ANOVA followed by Dunnett's multiple comparison test, with 'No tag' as the reference sample (*$p < 0.05$; N.S.: not significant).

with Red1 (S1H Fig), a protein dispensable for heterochromatin formation (Fig 3C), these results suggest that the observed Pab2 nuclear foci are likely involved in selective RNA degradation, rather than in heterochromatin assembly.

To directly assess Pab2's association with heterochromatin, particularly at centromeres, we performed ChIP-qPCR analysis using Pab2-GFP fusion proteins. This analysis revealed a preferential interaction of Pab2 at centromeres (Fig 5F), providing strong evidence for its direct association with this heterochromatic region. Importantly, analysis of Pab2 mutants revealed that while Pab2ΔRRM, Pab2[RRM3A], and Pab2[RA] mutant proteins retained centromere binding, the Pab2[RRM3A-RA] mutant significantly lost this interaction (Fig 5F). Taken together, these results demonstrate that Pab2 preferentially associates with centromeres in both RRM- and IDR-dependent manners, as only simultaneous disruption of both domains abolishes binding.

## Red5 and Rmn1, which interact with Pab2, are essential for constitutive heterochromatin assembly

To determine how Pab2 facilitates constitutive heterochromatin formation, we sought to identify Pab2-binding proteins using TurboID, a proximity-dependent protein labeling method [70]. Mass spectrometry analyses of biotinylated protein-enriched fractions revealed several proteins specifically enriched in the Pab2-TurboID sample (S9 Table). Among these, we focused on two putative RNA-binding proteins: the ZC3H3 ortholog Red5 (SJAG_00456) and the RBM26·27 ortholog Rmn1 (SJAG_03939). While orthologs of these proteins associate with Pab2 in *S. pombe* [16,31,38], they are dispensable for constitutive heterochromatin assembly in *S. pombe* [31]. We sought to determine their role(s) in *S. japonicus*.

Yeast two-hybrid assays (Y2H) confirmed that Pab2 interacts with both Red5 and Rmn1, and Red5 interacts with Rmn1 (Fig 6A and 6B). This suggests potential interactions that could form a trimeric complex. Growth on selective media (CM–Leu–Trp–His and CM–Leu–Trp–His–Ade) indicated that Pab2-Rmn1 and Red5-Rmn1 interactions are strong, whereas the Pab2 and Red5 interaction is relatively weak (Fig 6A and 6B). Consistent with these Y2H results, microscopic analyses showed that Pab2 colocalizes with both Red5 and Rmn1 (Fig 6C). Thus, we propose that Pab2 directly associates with Red5 and Rmn1.

To determine the roles of Red5 and Rmn1 in constitutive heterochromatin formation in *S. japonicus*, we generated a *red5–1* mutant and a *rmn1Δ* strain. Given that Red5 is essential in *S. pombe*, we isolated the *red5–1* mutant in *S. japonicus* while avoiding potential lethality. The *red5–1* mutant strain carries two amino acid substitutions (Gln253Arg and Thr265Ala) in the zinc-finger domain (S6A Fig), resulting in increased sensitivity to cold temperatures (S6B Fig). H3K9me2 ChIP-qPCR analysis revealed reduced levels of H3K9me2 at centromeres in *red5–1* and *rmn1Δ* (Fig 6D and 6E). Moreover, there was no evident cumulative effect when we combined either *red5–1* or *rmn1Δ* with *pab2Δ* (Fig 6D and 6E). Consistent with the decreased H3K9me2 levels, both *red5–1* and *rmn1Δ* were sensitive to TBZ (S6C Fig) and exhibited

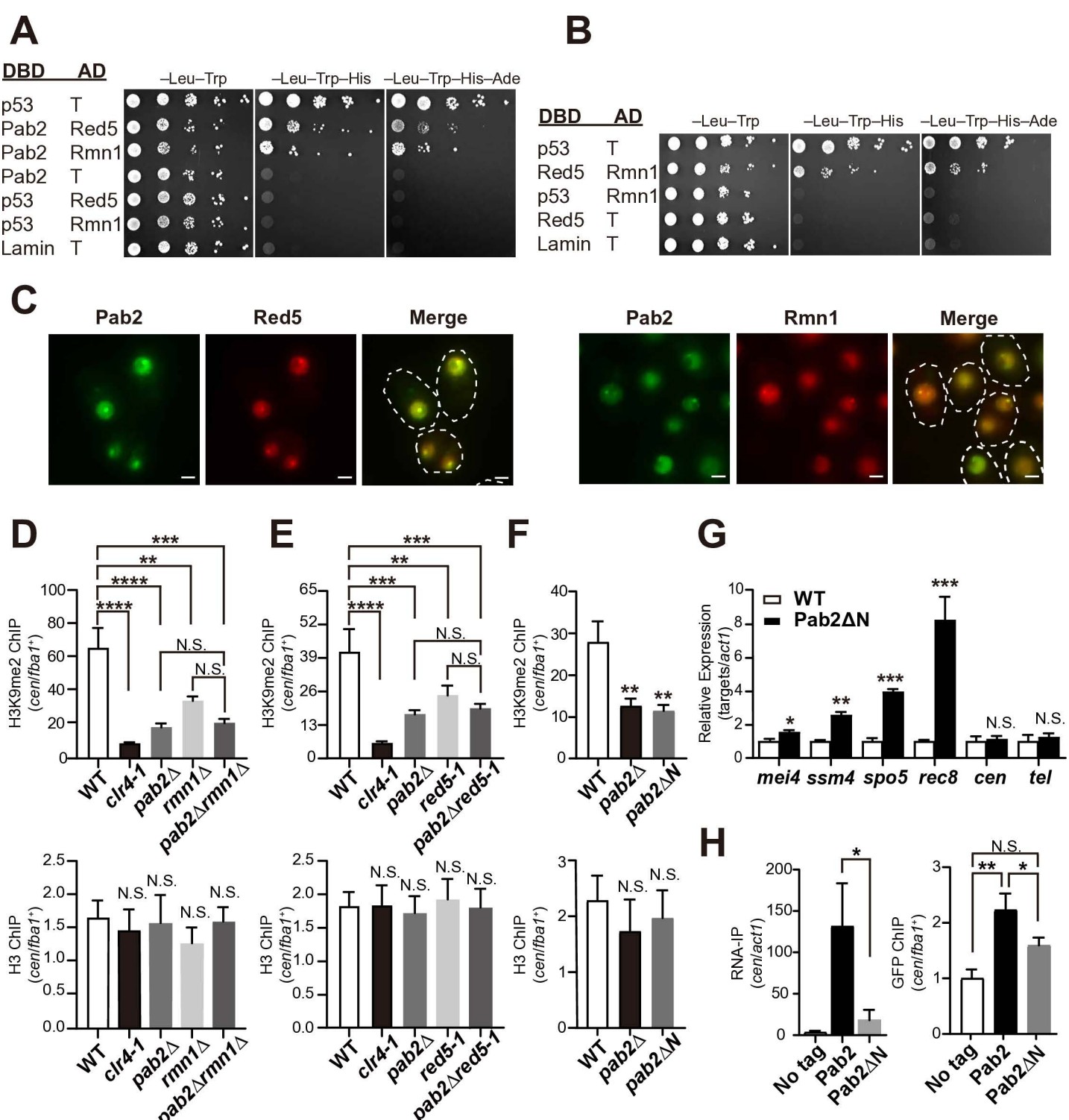

**Fig 6. Two Pab2-binding partners, Red5 and Rmn1, are required for centromeric heterochromatin assembly.** (A and B) Yeast two-hybrid assays examining interactions between Pab2-Rmn1, Pab2-Red5, and Red5-Rmn1. Ten-fold serial dilutions of *S. cerevisiae* AH109 strains carrying indicated plasmids were spotted onto minimal medium plates (CM) lacking leucine and tryptophan (−Leu–Trp), leucine, tryptophan, and histidine (−Leu–Trp–His), or leucine, tryptophan, histidine, and adenine (−Leu–Trp–His–Ade). The tumor suppressor p53 and T antigen (T) served as positive controls (p53-T), while Lamin and T antigen served as negative controls (Lamin-T). DBD, DNA-binding domain; AD, activation domain. (C) Fluorescent microscopy images showing the colocalization of Pab2 with Red5 and Rmn1. The white dotted lines indicate cell shapes. Scale bars, 2 μm. (D–F) ChIP-qPCR analysis of H3K9me2 and total H3 enrichment at centromeres using the indicated strains. Data are presented as mean ± SD from three independent experiments. Statistical significance was determined using a one-way ANOVA followed by Dunnett's multiple

comparison test, with WT as the reference sample (**$p < 0.01$, ***$p < 0.001$, and ****$p < 0.0001$; N.S.: not significant). (G) RT-qPCR analysis of transcripts derived from $mei4^+$, $ssm4^+$, $spo5^+$, and $rec8^+$, as well as centromeric ($cen$) and telomeric ($tel$) transcripts, in WT and $pab2\Delta$N strains. The transcripts were normalized to $act1$ mRNA to determine their relative expression levels. Data are presented as mean ± SD from three independent experiments. Statistical significance was determined using a two-tailed unpaired $t$-test (*$p < 0.05$, **$p < 0.01$, and ***$p < 0.005$; N.S.: not significant). (H) RNA-IP and ChIP-qPCR analysis of Pab2-GFP and Pab2$\Delta$N-GFP. (Left) RNA-IP followed by RT-qPCR to assess RNA-binding activity in cells expressing untagged Pab2 (No tag), Pab2-GFP, or Pab2$\Delta$N-GFP. Centromeric RNA ($cen$) was normalized to $act1$ mRNA to determine their relative enrichment. Data are presented as mean ± SD from three independent experiments. Statistical significance was determined using a two-tailed unpaired $t$-test, with 'Pab2' as the reference sample (*$p < 0.05$). (Right) ChIP-qPCR analysis to assess Pab2-GFP enrichment at centromeres using the indicated strains. Data are presented as mean ± SD from three independent experiments. Statistical significance was determined using a one-way ANOVA followed by Dunnett's multiple comparison test, with 'No tag' as the reference sample (*$p < 0.05$ and **$p < 0.01$; N.S.: not significant).

increased levels of centromeric transcripts (S6D Fig). These results strongly suggest that Red5 and Rmn1 cooperate with Pab2 to assemble constitutive heterochromatin.

To further investigate the significance of Pab2's interactions with Red5 and Rmn1 in heterochromatin assembly, we sought to determine the functional consequences of disrupting these interactions. Y2H assays using three Pab2 fragments (N-terminal domain, RRM, and IDR) revealed that the N-terminal domain of Pab2 is responsible for binding to Red5 and Rmn1 (S6E Fig), leading us to construct a strain that expresses Pab2 lacking the N-terminal domain ($pab2\Delta$N). Western blotting confirmed that Pab2$\Delta$N-GFP expression levels were comparable to those of Pab2-GFP (S6F Fig). While Pab2$\Delta$N-GFP failed to form nuclear foci, Red5 and Rmn1 still formed nuclear dots (S6G Fig), indicating that Red5 and Rmn1 localization does not depend on their interactions with Pab2. ChIP-qPCR analysis revealed reduced H3K9me2 levels at centromeres in $pab2\Delta$N cells (Fig 6F), and $pab2\Delta$N cells displayed elevated sensitivity to TBZ (S6H Fig), indicating a significant loss of heterochromatin. Furthermore, transcripts derived from $mei4^+$, $ssm4^+$, $spo5^+$, and $rec8^+$ were increased in $pab2\Delta$N cells (Fig 6G). Intriguingly, Pab2$\Delta$N-GFP exhibited a substantial loss of RNA- and centromere-binding ability (Fig 6H). This suggests that the N-terminal domain of Pab2 plays a crucial role in its RNA and centromere binding. Further investigation is needed to determine whether this is due to direct binding, structural contributions, or modulation of interactions with Red5/Rmn1. Taken together, these findings demonstrate that the interaction between Pab2 and Red5/Rmn1 is essential for heterochromatin assembly.

## The canonical poly(A) polymerase Pla1 is essential for the formation of constitutive heterochromatin

Studies in *S. pombe* have shown that centromeric transcripts are transcribed by RNA polymerase II (Pol II), suggesting that these transcripts are polyadenylated [43] and the Pol II C-terminal domain recruits the canonical poly(A) polymerase Pla1 to add poly(A) tail [71]. Given that poly(A) tails are generally recognized by poly(A)-binding proteins [72], we hypothesized that *S. japonicus* Pla1 (SJAG_02720) might indirectly contribute to heterochromatin assembly by facilitating Pab2 recruitment to these transcripts through polyadenylation. To test this, we analyzed a *pla1* mutant strain (*pla1–3*), which contains multiple amino acid substitutions in both the N-terminal nucleotidyltransferase domain and the C-terminal RNA-binding domain (S10 Table). The *pla1–3* mutant strain exhibited increased sensitivity to both low and high temperatures (Fig 7A). Consistent with a potential role for Pla1 in heterochromatin formation, ChIP-qPCR analysis indicated that H3K9me2 levels at centromeres were reduced in *pla1–3* cells (Fig 7B). Additionally, *pla1–3* cells showed greater sensitivity to TBZ than WT cells (Fig 7C). Moreover, RT-qPCR analysis demonstrated increased levels of transcripts derived from $mei4^+$, $spo5^+$, $rec8^+$, and centromeres in *pla1–3* (Fig 7D). Finally, yeast two-hybrid assays did not reveal a direct physical interaction between Pla1 and Pab2 (Fig 7E),

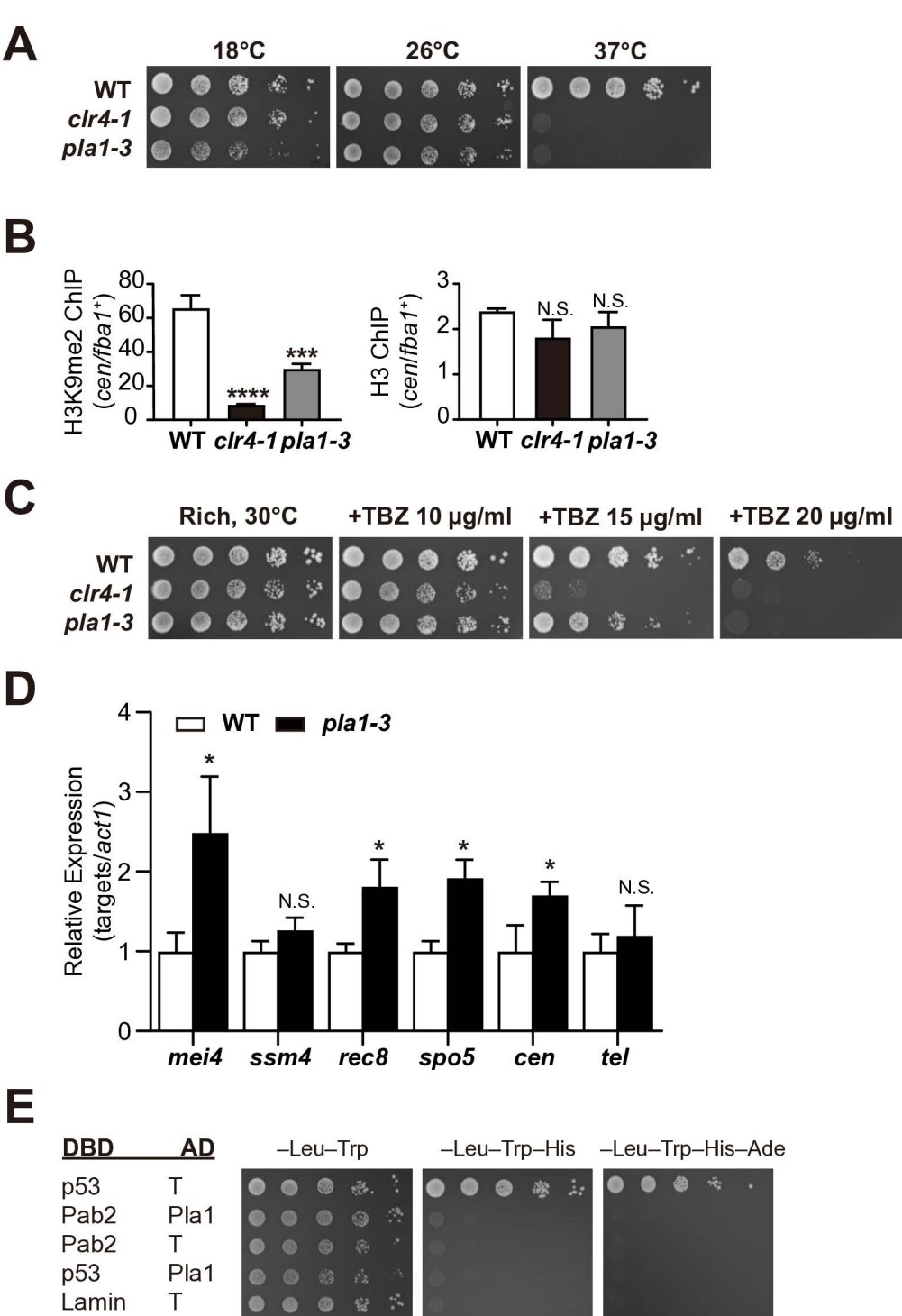

**Fig 7. Pla1 is required for centromeric heterochromatin.** (A) Growth assay of wild-type (WT), *clr4–1* and *pla1–3* cells. Ten-fold serial dilutions were spotted onto complete medium plates and incubated at the indicated temperatures. (B) ChIP-qPCR analysis of H3K9me2 and total H3 levels at centromeres in the indicated strains. Data are presented as mean ± SD from three independent experiments. Statistical significance was determined using a one-way ANOVA followed by Dunnett's multiple comparison test, with WT as the reference sample (***$p < 0.001$ and ****$p < 0.0001$; N.S.: not significant). (C) Thiabendazole (TBZ) sensitivity assay of WT, *clr4–1*, and *pla1–3* cells. Ten-fold serial dilutions were spotted onto complete medium plates with or without TBZ and grown at 30°C. (D) RT-qPCR analysis of transcripts from *mei4+*, *ssm4+*, *spo5+*, and *rec8+*, as well as centromeric (*cen*) and telomeric (*tel*) transcripts, in WT and *pla1–3*. The transcripts were normalized to *act1* mRNA to determine their relative expression

levels. Data are presented as mean ± SD from three independent experiments. Statistical significance was determined using a two-tailed unpaired *t*-test (*$p < 0.05$; N.S.: not significant). (E) Yeast two-hybrid assays examining the interaction between Pab2 and Pla1. Ten-fold serial dilutions of *S. cerevisiae* AH109 strains carrying indicated plasmids were spotted onto minimal medium plates (CM) lacking leucine and tryptophan (–Leu–Trp), leucine, tryptophan, and histidine (–Leu–Trp–His), or leucine, tryptophan, histidine, and adenine (–Leu–Trp–His–Ade). The tumor suppressor p53 and T antigen (T) served as positive controls (p53-T), while Lamin and T antigen served as negative controls (Lamin-T). DBD, DNA-binding domain; AD, activation domain.

suggesting that Pab2 and Pla1 likely function in distinct steps of the heterochromatin assembly pathway. These findings suggest a possible link between Pla1 function and constitutive heterochromatin formation in *S. japonicus*, potentially through its role in polyadenylation of centromeric transcripts and subsequent influence on Pab2 recruitment. Further investigation is required to fully elucidate the role of Pla1 in heterochromatin assembly.

## Dri1 is dispensable for constitutive heterochromatin

Dri1 (SJAG_02381) was identified as a Pab2-proximal protein in our TurboID experiments (S9 Table). Dri1 is an RNA-binding protein with an RRM, and recent reports have demonstrated its involvement in heterochromatin assembly in *S. pombe* [73,74]. Yeast two-hybrid assays indicated slightly better growth of yeast cells carrying DBD-Rmn1 and AD-Dri1 on –Leu–Trp–His plates than those carrying DBD-p53 and AD-Dri1 (S7A Fig). This result suggests a possible, but weak, interaction between Rmn1 and Dri1. However, due to the qualitative nature of the assay, our observation does not provide sufficient evidence to confirm a direct and biologically relevant interaction between Rmn1 and Dri1. Further analyses are required to determine if there is any significant interaction. In addition, ChIP-qPCR analysis demonstrated that H3K9me2 levels at centromeres and telomeres in WT and *dri1Δ* were comparable (S7B Fig). Furthermore, *dri1Δ* cells were not sensitive to TBZ (S7C Fig). These results indicate that *S. japonicus* Dri1 is dispensable for centromeric and telomeric H3K9me2, highlighting differences in heterochromatin assembly mechanisms between *S. japonicus* and *S. pombe*.

## Ppn1/PPP1R10, a CPSF constituent, interacts with Pab2 and the H3K9 methyltransferase Clr4/SUV39

Our TurboID approach also identified Ppn1 (SJAG_03500), the ortholog of PPP1R10 and a component of CPSF complex, as a Pab2-associated protein (S9 Table). In *S. pombe*, Ppn1 is part of the DPS (Dis2-Ppn1-Swd22) module within the CPSF complex [75]. While the involvement of CPSF in heterochromatin formation is established in *S. pombe* [26,27], the specific contribution of CPSF to this process remains unclear. To investigate the role of Ppn1 in *S. japonicus* heterochromatin assembly, we first examined its interaction with Pab2. Yeast two-hybrid assays confirmed the interaction between Pab2 and Ppn1 (Fig 8A). Notably, none of the Pab2 fragments (N, RRM, IDR, ΔN, ΔRRM, or ΔIDR) interacted with Ppn1 (S8A Fig), indicating that the full-length Pab2 is required for this interaction. Consistent with these results, fluorescent microscopy demonstrated that Ppn1 foci colocalize with Pab2 foci (Fig 8B). Furthermore, Ppn1 nuclear foci are largely absent in the absence of Pab2 (S8B Fig), while western blotting confirmed that Ppn1 expression levels in *pab2Δ* was comparable to those in WT (S8C Fig). These results further support a physical association between Pab2 and Ppn1. Remarkably, we found that Ppn1 and Rmn1 also directly interact with Clr4, as confirmed by yeast two-hybrid assays (Fig 8C). We also found that the Clr4 middle domain (amino acids: 76–220), but not the chromodomain (amino acids: 1–75) or SET domain (amino acids:

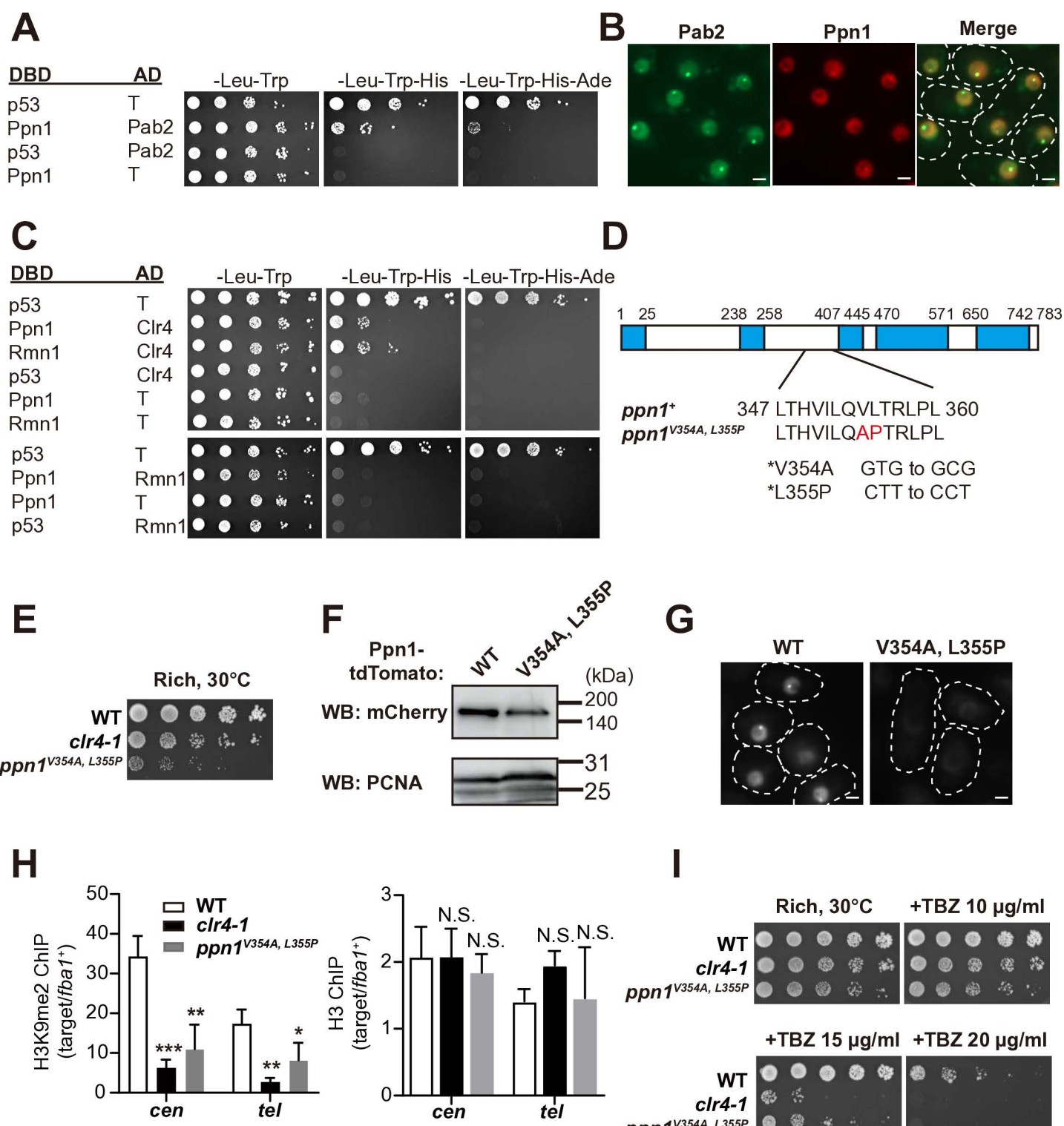

**Fig 8. The CPSF constituent Ppn1 is required for the assembly of constitutive heterochromatin.** (A) Yeast two-hybrid assays assessing the interaction between Pab2 and Ppn1. Ten-fold serial dilutions of *S. cerevisiae* AH109 strains carrying the indicated plasmids were spotted onto selective medium plates lacking leucine and tryptophan (–Leu–Trp), leucine, tryptophan, and histidine (–Leu–Trp–His), or leucine, tryptophan, histidine, and adenine (–Leu–Trp–His–Ade). p53-T antigen (T) and Lamin-T served as positive and negative controls, respectively. (B) Localization of Pab2-GFP and Ppn1-tdTomato was examined by fluorescent microscopy. The white dotted lines indicate cell shapes. Scale bars, 2 μm. (C) Yeast two-hybrid assays testing Ppn1-Clr4, Rmn1-Clr4, and Ppn1- Rmn1 interactions were performed as described in (A). (D) Schematic of Ppn1 showing the V354A and L355P substitutions, which disrupt the interaction with Clr4. Blue boxes indicate predicted

intrinsically disordered regions. (E) Growth assay of wild-type (WT), *clr4–1*, and the *ppn1*$^{V354, L355P}$ mutant cells. Ten-fold serial dilutions were spotted onto complete medium plates, and the plates were incubated at 30°C. (F) Western blotting of Ppn1- and Ppn1$^{V354, L355P}$-tdTomato, with Pcn1/PCNA as a loading control. (G) Localization of Ppn1- and Ppn1$^{V354, L355P}$-tdTomato in vegetative cells. The white dotted lines indicate cell shapes. Scale bars, 2 μm. (H) Levels of H3K9me2 and total H3 at centromeres and telomeres in WT, *clr4–1*, and *ppn1*$^{V354, L355P}$ mutant strains assessed by ChIP-qPCR. Data are presented as mean ± SD from three independent experiments. Statistical significance was determined using a one-way ANOVA followed by Dunnett's multiple comparison test, with WT as the reference sample (*$p < 0.05$, **$p < 0.01$, and ***$p < 0.001$; N.S.: not significant). (I) Thiabendazole (TBZ) sensitivity assay of WT, *clr4–1*, and *ppn1*$^{V354, L355P}$ mutant cells. Ten-fold serial dilutions were spotted onto complete medium plates with or without TBZ and grown at 30°C.

221–518), interacts with Ppn1 (S8D Fig). These findings suggest that Clr4, Pab2, and Ppn1 may function together within heterochromatic domains.

## Mutations in Ppn1 disrupt constitutive heterochromatin

To investigate Ppn1's role in heterochromatin formation in *S. japonicus*, we initially attempted to generate a *ppn1Δ* strain. Although *ppn1*$^{+}$ is non-essential for viability in *S. pombe* [76], we were unable to obtain a viable *ppn1Δ* strain in a wild-type *S. japonicus* strain. This observation, while requiring further investigation, hints at a potential difference in Ppn1 function or genetic context between the two species, possibly involving RNAi-dependent heterochromatin assembly as suggested by previous reports on RNAi components in *S. japonicus* [24].

Because we could not obtain a *ppn1Δ* strain, we next aimed to generate *ppn1* mutant strains using PCR-based random mutagenesis [77]. Despite repeated attempts, we could not isolate any temperature- or cold-sensitive *ppn1* mutants. Therefore, to specifically investigate the role of the Pab2-Ppn1 interaction in heterochromatin assembly, we employed a yeast reverse two-hybrid system [78] to select for *ppn1* mutations that disrupt this interaction. This screen isolated a *ppn1* mutant with two amino acid substitutions: Val354Ala and Leu355Pro (Fig 8D). According to AlphaFold predictions [79,80], these substitutions are located within an α-helix of Ppn1, and the proline residue (Leu355Pro) likely disrupts the helix structure [81]. Standard yeast two-hybrid assay showed that Ppn1$^{V354A, L355P}$ lost its ability to interact with Clr4 and Pab2 (S8E Fig). Introduction of the V354A and L355P mutations into the endogenous *ppn1*$^{+}$ gene resulted in a significant growth defect (Fig 8E). Western blotting revealed that the mutations reduced the steady-state level of Ppn1 (Fig 8F). Although Ppn1$^{V354A, L355P}$-tdTomato is detectable by western blotting, we did not observe any nuclear condensates (Fig 8G). This lack of localization, along with the reduced Ppn1 protein level (Fig 8F), suggests that the mutations disrupt Ppn1 localization and/or stability. In addition, ChIP-qPCR analysis showed decreased H3K9me2 levels at centromeres and telomeres in *ppn1*$^{V354, L355P}$ mutant cells (Fig 8H). The *ppn1*$^{V354, L355P}$ mutant also exhibited increased sensitivity to TBZ (Fig 8I), and transcripts derived from *mei4*$^{+}$, *rec8*$^{+}$, and *spo5*$^{+}$, and centromeres accumulated in the mutant cells (S8F Fig). These findings suggest that Ppn1 contributes to heterochromatin assembly, selective gene silencing, and maintenance of genome stability in *S. japonicus*.

## Discussion

In this study, we characterized the nuclear poly(A)-binding protein Pab2/PABPN1 in *S. japonicus* and demonstrated that Pab2 cooperates with two putative RNA-binding proteins (Red5 and Rmn1) and CPSF to promote constitutive heterochromatin assembly. We also showed that both the Pab2 RNA-binding domain and the intrinsically disordered region are essential for Pab2 condensate formation and for heterochromatin formation. Our results provide insights into Pab2's functions and highlight the importance of nuclear condensates in this process. Surprisingly, our findings reveal contrasting mechanisms of constitutive heterochromatin assembly in two *Schizosaccharomyces* species: Pab2-dependent in *S. japonicus* and Pab2-independent in *S. pombe*. Furthermore, consistent with previous reports [21,24,82], our

findings underscore that *S. japonicus* possesses unique characteristics that make it suitable for studying chromosome biology.

## Divergence between *S. japonicus* and *S. pombe*

The finding that Pab2 is essential for proper growth, efficient mating, and constitutive heterochromatin formation in *S. japonicus* represents a significant divergence from its role in *S. pombe*. While *S. pombe* Pab2 plays a role in suppressing meiotic gene expression [33,36], it does not seem essential for normal vegetative growth or efficient mating [33]. Furthermore, to the best of our knowledge, no prior studies have clearly demonstrated a role for Pab2 in the assembly of constitutive heterochromatin in *S. pombe*. This functional divergence suggests that Pab2 has acquired novel roles in *S. japonicus*. Several factors could contribute to these differences, including variations in genome organization [21] and/or differences in the mechanisms for cell growth and meiosis. These possible differences between the two species could also necessitate a more critical role for Pab2 in *S. japonicus*.

This divergence in Pab2 function is further exemplified by comparisons of heterochromatin assembly. In *S. pombe,* none of Red1, Red5, or Rmn1 is essential for H3K9 methylation at constitutive heterochromatic domains [31]. In contrast, Pab2, Red5, and Rmn1 are all required for centromeric heterochromatin in *S. japonicus* (Fig 3 and 6). Notably, we also found that Mit1 plays a critical role in H3K9 methylation at centromeres in *S. japonicus* (Fig 4), though not observe in *S. pombe* [15]. These differences in heterochromatin assembly mechanisms likely stem from variations in centromeric DNA sequences. *S. japonicus* centromeric DNA contains Ty3/Gypsy-type retrotransposons [21] and utilizes RNAi and heterochromatin for retrotransposon suppression [24]. Transposable elements are found at the centromeres of various eukaryotes [83–85], and RNAi acts as a defense system against them [86,87]. *S. pombe*, however, lacks such transposable elements within centromeres, instead using repetitive sequences called *dg*/*dh* [88]. This suggests that Pab2 in *S. japonicus* targets these centromeric retrotransposons. This idea can be supported by the involvement of Pab-1, a PABP in *C. elegans*, in transposon silencing [89] and the known connections between retrotransposons, the poly(A) tail, and PABPs [90–93]. It would be of interest to investigate the role of PABPs in heterochromatin assembly and transposon suppression in multicellular organisms.

Beyond centromeric structure, other differences exist between two species. Mitotic condensed chromosomes are visible in *S. japonicus* but not in *S. pombe*, and *S. japonicus* cells are larger [82]. These characteristics, along with the unique aspects of Pab2, highlight the value of *S. japonicus* as a model organism for uncovering novel aspects of conserved cellular processes not readily apparent in S. pombe. s its unique strength in studying cell biology and chromosome biology. While *S. pombe* research infrastructure is more developed, resources for *S. japonicus* are growing, including two programs we have developed [94], and we anticipate further development as the advantages of this organism are recognized.

## Dissecting Pab2 functional domains

Pab2 comprises three domains: the N-terminal domain, the RNA recognition motif (RRM), and the C-terminal intrinsically disordered region (IDR). Yeast two-hybrid assays indicated that the N-terminal domain interacts with Red5 and Rmn1 (S6E Fig), suggesting its importance in forming the Pab2-Red5-Rmn1 complex. Consistent with this, deletion of the N-terminal domain (Pab2ΔN) abolished nuclear dot formation and resulted in a substantial loss of centromeric RNA binding and reduced association with centromeres (Fig 6). These findings demonstrate that Red5 and Rmn1 are essential for the stable association of the Pab2

complex with centromeric RNA and heterochromatin, suggesting that the Pab2-Red5-Rmn1 complex, rather than Pab2 alone, is the primary functional unit at heterochromatic domains.

The RRM is a well-characterized RNA-binding domain [68], and we hypothesized that Pab2's association with centromeric transcripts via its RRM is critical for constitutive heterochromatin formation. Mutational analysis of the RRM strongly supports this hypothesis. The Pab2$^{RRM3A}$ mutant, which exhibited a substantial reduction in RNA-binding ability with detectable RNA binding, failed to form nuclear condensates and showed reduced levels of H3K9me2 at centromeres. This demonstrates that a critical threshold of RNA binding is necessary for Pab2 to form functional condensates and promote heterochromatin assembly. This heterochromatin defect, resulting from even partial disruption of RNA binding, highlights the importance of the Pab2-centromeric RNA interaction and suggests that centromeric RNAs may act as a structural component of the condensates, scaffolding the assembly of droplets containing heterochromatin factors as described previously [95,96].

IDRs are known to be crucial for biomolecular condensate formation through phase separation [54,55]. Our analyses demonstrated that the Arg residues within the IDR, which are crucial for the IDR interactions that assemble biological condensates [62], are essential for Pab2 droplet formation both *in vitro* and *in vivo* (Fig 4). RNA-IP data showed that Pab2$^{RRM3A}$ still possess some RNA-binding ability, while Pab2$^{RRM3A-RA}$ appears to have lost the ability completely. This indicates that the IDR also contributes independently to the association with centromeric RNAs, consistent with previous findings that Arg/Gly-rich (GAR) domains and RGG/RG boxes can bind nucleic acids [97]. Therefore, both RRM and IDR are involved in RNA association within Pab2 nuclear condensates.

## Phase separation and heterochromatin assembly

The involvement of RNA-binding proteins and phase separation in heterochromatin assembly is increasingly recognized across diverse eukaryotes. In mammals, for example, RNA-binding proteins like FUS and hnRNP U have been implicated in heterochromatin formation [98,99], and phase separation of HP1 and other heterochromatin factors has been shown to be crucial for heterochromatin domain organization [61,100]. Our findings in *S. japonicus* further support these conserved principles, highlighting the importance of RNA and phase separation in heterochromatin regulation. However, while some aspects are conserved, the specific proteins involved and their precise roles can differ significantly between species. For example, while Pab2, Red5, and Rmn1 play a crucial role in *S. japonicus* heterochromatin assembly, their orthologs have a less critical role in *S. pombe*. This underscores the importance of studying diverse model organisms to fully understand the complexity of heterochromatin regulation.

## Differential roles of Pab2 condensates

Our observation that both the RRM and IDR of Pab2 are required for the formation of nuclear condensates raises the question of how these structures contribute to Pab2's diverse functions. Notably, the visible Pab2 condensates colocalize with Red1 (S1H Fig), a protein involved in selective mRNA decay but dispensable for heterochromatin assembly (Fig 3C). This suggests a role for these condensates in the mRNA degradation. Importantly, Pab2 condensates do not colocalize with Chp1 foci, which mark heterochromatic domains (S5C Fig). However, ChIP-qPCR data demonstrate that Pab2 also associates with centromeric regions (Fig 5F), suggesting a distinct interaction with heterochromatin, likely in a less concentrated manner. Given the essential role of centromeric RNA binding for heterochromatin assembly as demonstrated by our RRM mutant analysis (Fig 5), we propose that Pab2 can form distinct types of condensates with different properties: one type primarily involved in selective mRNA decay and

another, dependent on centromeric RNAs, involved in heterochromatin assembly. Specifically, we hypothesize that the presence of centromeric RNAs alters the composition and/or properties of Pab2 condensates, enabling the recruitment or stabilization of key heterochromatin factors such as Clr4 and Mit1 at centromeric regions. This functional diversification of Pab2, mediated by distinct condensate properties, suggest a complex interplay between RNA metabolism and heterochromatin regulation.

## Possible roles of Red5 and Rmn1 in heterochromatin assembly

While Red5 and Rmn1 associate with Pab2 in both *S. japonicus* (this study) and *S. pombe* [16,17,31], they are essential for proper heterochromatin assembly only in *S. japonicus* (Fig 6). However, their precise molecular functions in *S. japonicus* remain to be determined. It is possible that Red5 and Rmn1 are essential for stabilizing the association of the Pab2 complex with centromeric RNAs and heterochromatin, as discussed above. Furthermore, given that Pab2 exhibits preferential binding to putative meiotic mRNAs and centromeric RNAs (Fig 2D) despite its general affinity for poly(A) RNA, Red5 and Rmn1 could contribute to the RNA-binding specificity of the Pab2 complex. It is also plausible that Rmn1 (and potentially Red5) facilitates the recruitment or stabilization of other heterochromatin factors, such as Clr4 and Mit1, at heterochromatic domains. Finally, considering that Red5 and Rmn1 also contain intrinsically disordered regions and form nuclear foci independently of Pab2's N-terminal domain (S6G Fig), they could contribute to the formation, stability, or organization of Pab2 condensates themselves. Future studies, such as co-immunoprecipitation experiments to examine the interactions of Red5/Rmn1 with other heterochromatin factors and RNA-binding assays using specific RNA sequences, will be crucial for elucidating the precise molecular functions of Red5 and Rmn1.

## Ppn1 and its potential connection to heterochromatin assembly

Previous studies have implicated the CPSF complex in the formation of heterochromatin in *S. pombe* [26,27], but its role in heterochromatin assembly is less clear. Our finding that the CPSF component Ppn1 interacts with Clr4 raises the possibility that Ppn1 may also have a function at heterochromatic loci. Given the interaction of Ppn1 with Pab2 (Fig 8), we hypothesize that Ppn1 might modulate the function of Pab2 at these loci. For example, Ppn1 could influence the efficiency of Pab2 condensate formation or stability. Alternatively, Ppn1 could attract Clr4 and Mit1 to Pab2 condensates. However, our data only demonstrate interactions between Ppn1, Pab2, and Clr4, highlighting the need for further experiments to characterize Ppn1 and determine whether the entire CPSF complex is involved.

## A model for Pab2-mediated heterochromatin assembly in *S. japonicus*

Based on our findings, we propose a working model for Pab2-mediated heterochromatin assembly at centromeres in *S. japonicus* (Fig 9). This model centers on the role of centromeric non-coding RNAs (ncRNAs) as a scaffold for heterochromatin formation. The Pab2 complex is recruited to heterochromatic domains via these ncRNAs. The RRM provides a crucial threshold of RNA binding, and the IDR further strengthens this interaction. Subsequent phase separation of Pab2, driven by IDR interactions, leads to the formation of nuclear condensates at heterochromatic domains. The bound ncRNAs likely become integral structural components of these condensates, contributing to their specific properties. This concept of ncRNAs acting as scaffolds for heterochromatin assembly is supported by a previous study in mammalian systems, where Suv39h (a mouse ortholog of Clr4) has been shown to directly bind ncRNAs derived from heterochromatic regions, leading to stabilized Suv39h localization

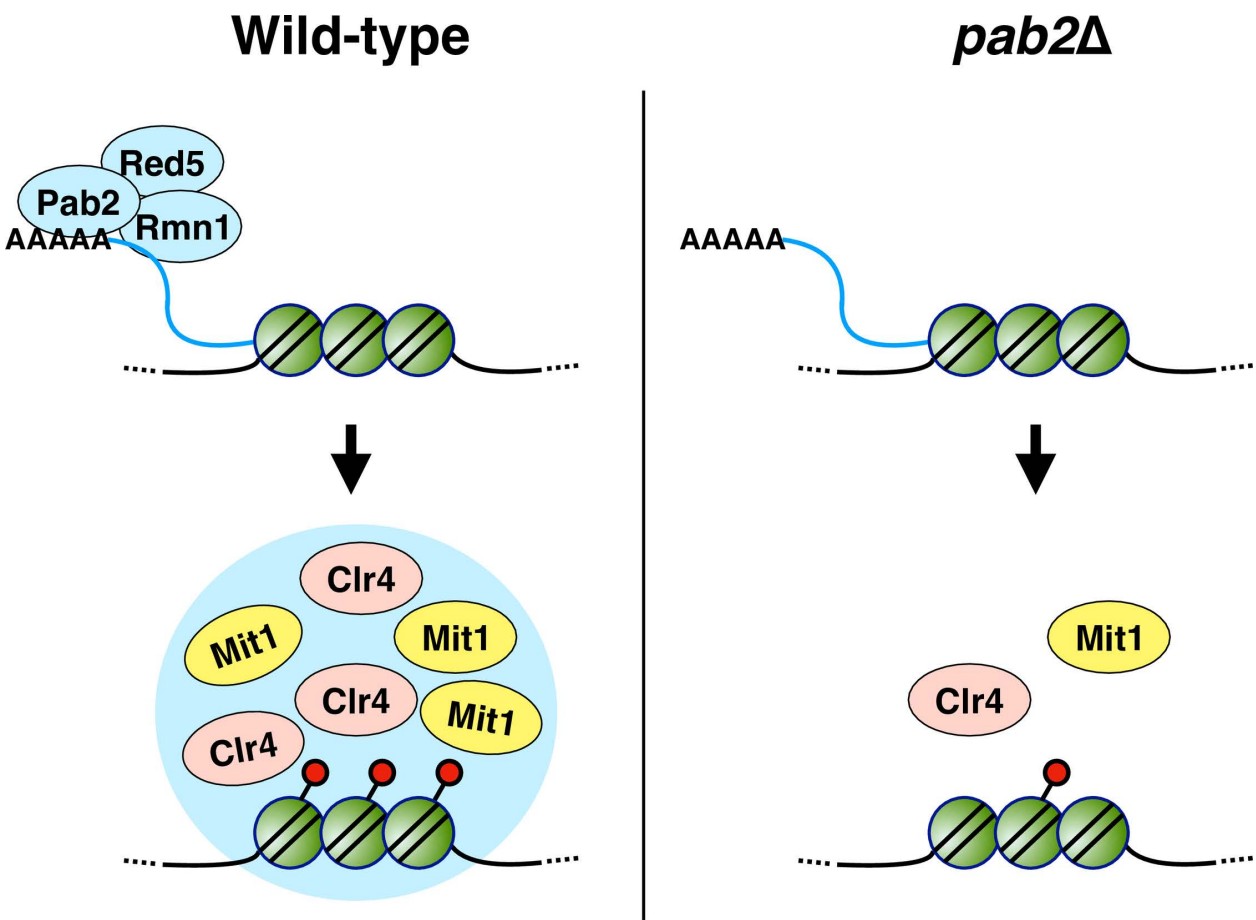

**Fig 9. A model for Pab2-mediated heterochromatin assembly in *S. japonicus*.**

at pericentromeric heterochromatin [101]. Pab2 condensates then promote heterochromatin assembly by concentrating heterochromatin factors such as Clr4 and Mit1 at these loci, thereby facilitating their activity. Moreover, Ppn1, by interacting with both Pab2 and Clr4, may coordinate their functions, potentially by stabilizing the interaction between Pab2 and Clr4. This working model highlights the interplay between ncRNA, phase separation, and protein-protein interactions in Pab2-mediated heterochromatin assembly.

The Pab2 complex (Pab2, Red5, and Rmn1) is recruited to heterochromatic regions via non-coding RNAs (ncRNAs) derived from these loci. The Pab2 complex then undergoes phase separation, forming nuclear condensates. These condensates concentrate key heterochromatin factors, such as the H3K9 methyltransferase Clr4 and the chromatin remodeler Mit1, at heterochromatic domains. This concentration facilitates their recruitment and/or stabilizes the association of these factors, thereby promoting heterochromatin assembly. In the absence of Pab2, heterochromatin assembly is significantly impaired.

## Future directions

This study raises several important questions that warrant further investigation, both in *S. japonicus* and in other systems. First, directly testing the interaction between Pab2 and RNAi components in *S. japonicus* would be important to establish a concrete link between these pathways, given the established role of RNAi in heterochromatin formation in this

organism. Second, investigating the *in vivo* dynamics of Pab2 condensates and their relationship to heterochromatin domains will further elucidate the mechanism of heterochromatin assembly. Third, determining the precise roles of Red5 and Rmn1 within the Pab2 complex, and exploring their potential interactions with other heterochromatin factors, such as those involved in histone modification, will also be valuable. Fourth, characterizing Mit1 function in constitutive heterochromatin assembly, which is not evident in *S. pombe*, will advance our understanding of the heterochromatin assembly pathway in *S. japonicus*. Finally, further studies are needed to investigate the role of polyadenylation, including the contribution of Pla1, in heterochromatin assembly and the recruitment of associated factors.

Given our findings demonstrate a role of Pab2 in heterochromatin formation in *S. japonicus*, it will be crucial to determine: (1) whether the functional interplay between PABP and heterochromatin assembly observed in *S. japonicus* is conserved in other species, (2) whether human PABPs similarly interact with factors involved in heterochromatin formation or the RNAi pathway, and (3) whether the principles governing the formation and function of Pab2 condensates found in *S. japonicus* are applicable to the organization of heterochromatin in more complex genomes. Addressing these questions will deepen our understanding of heterochromatin regulation and potentially reveal novel roles for PABPs beyond mRNA metabolism.

## Materials and Methods

### Media and *S. japonicus* strain construction

Strains were cultured in YES (Yeast Extract Supplement) rich medium. Gene deletion and C-terminal tagging strains of *S. japonicus* were generated using a PCR-based gene targeting method as described previously [102]. Primers for the PCR-based strain constructions were designed using Yesprit [94]. Specific point mutations (Pab2$^{RA}$, Pab2$^{RRM3A}$, Pab2$^{RRM3A-RA}$, and Ppn1$^{V354A, L355P}$) were introduced using the "pop-in, pop-out" allele replacement method [103]. Transformation of *S. japonicus* was performed as described previously [104]. DNA fragments used for transformation were amplified using PrimeSTAR Max DNA Polymerase (Takara Bio Inc.). Transformants were selected on YES plates containing an antibiotic or on Pombe minimal glutamate (PMG) medium plates lacking uracil. Correct transformants were confirmed by genomic PCR using 2 × Taq Master Mix (Novoprotein Scientific Inc.).

Temperature/cold-sensitive (ts/cs) mutant strains were generated using an error-prone PCR-based random mutagenesis method [77]. Briefly, a target gene (*clr4⁺*, *red5⁺*, or *pla1⁺*), including the drug resistance marker *natMX6*, was subjected to error-prone PCR using LA Taq DNA Polymerase (Takara Bio Inc.). To increase the mutation rate, dGTP was added to the PCR reaction to a final concentration of 0.8 mM. The resulting PCR products were used to transform a wild-type strain, and the transformants were selected on YES plates containing the appropriate antibiotics at 26°C for 3–4 days. Single colonies were replica-plated onto YES plates containing Phloxine B (PhB) and incubated at 18°C, 26°C, and 37°C. PhB stains dead cells deep red, whereas viable cells appear pink. Transformants exhibiting growth defects at 18°C and/or 37°C were subjected to ChIP-qPCR analysis of H3K9me2 to identify mutants defective in heterochromatin assembly (resulting in *clr4–1*, *red5–1*, and *pla1–3*). The mutations in the corresponding genes of three mutants were determined by Sanger sequencing of genomic DNA. *S. japonicus* strains used in this study are listed in S11 Table.

### Western blotting

Protein extracts were prepared using previously described methods [39,105]. For detection of epitope tagged proteins, anti-GFP (No.11814460001, Roche), anti-c-myc (HT101–01,

Transgene), anti-HA (HT301–01, Transgene), or anti-mCherry (bsm-33131M, Bioss) antibody was used. The *S. japonicus* PCNA ortholog Pcn1 was detected using an anti-PCNA antibody (A13336, ABclonal), which served as a protein loading control.

## Fluorescent microscopy

Fluorescence and differential interference contrast (DIC) images were acquired using a Zeiss Axio Imager Z2 Upright Microscope (Carl Zeiss MicroImaging). ZEN lite 2012 software (Carl Zeiss MicroImaging) was used to process RAW images.

## RNA analyses

Yeast cells were harvested at mid log-phase ($OD_{600} = 0.5–1$). Total RNA was isolated using the MasterPure Yeast RNA Purification Kit (Lucigen). Strand-specific RNA-Seq was conducted using the VAHTS Universal V8 RNA-seq Library Prep Kit for Illumina (NR605–02, Vazyme) and NovaSeq 6000 S4 Reagent Kit (Illumina). Raw sequencing reads were processed as described previously [37]. The RNA-seq datasets generated in this study have been deposited in the Gene Expression Omnibus (GEO) under accession number GSE237964. Gene ontology (GO) enrichment analysis was performed using Metascape [106]. Due to the limited availability of *S. japonicus* gene annotations in Matascape, orthologous genes in *S. pombe* were used for this analysis. RT-qPCR was performed using the PrimeScript II 1st strand cDNA Synthesis Kit (Takara Bio Inc.) and ChamQ Universal SYBR qPCR Master Mix (Vazyme). Primers used for qPCR are listed in S12 Table.

## Chromatin Immunoprecipitation (ChIP)

ChIP was performed as described previously [37,107] with slight modifications. Specifically, fission yeast cells were grown to mid log-phase ($OD_{600} = 0.5–1$) at 26°C (permissive temperature for all mutant strains) and harvested. Cells were washed with $1 \times$ PBS and then fixed in 3% formaldehyde for 10 minutes at room temperature. Chromatin was sheared by sonication using a Bioruptor Plus sonication device (Diagenode) to an average size of less than 500 base pairs. For immunoprecipitation, 2 μg of antibodies against H3K9me2 (ab1220, Abcam), H3K9me3 (DF6938, Affinity Biosciences), histone H3 (ab1791, Abcam), or anti-GFP (ab290, Abcam) antibody as well as 20 μL of protein A/G magnetic beads (P2108, Beyotime Biotechnology) were used for each sample. After immunoprecipitation, samples were reverse crosslinked by incubation at 68°C overnight. Primer sequences used for ChIP-qPCR are listed in S12 Table.

## Identification of Pab2-binding proteins by TurboID and mass spectrometry

Proximity-dependent biotin labeling by TurboID in the search for Pab2-binding proteins was carried out as described previously [70]. Specifically, an *S. japonicus* strain expressing Pab2-TurboID and the corresponding parental strains were lysed using RIPA buffer [50 mM Tris–HCl (pH 7.5), 150 mM NaCl, 1.5 mM $MgCl_2$, 1 mM EGTA, 0.1% SDS, and 1% NP-40] supplemented with 0.4% sodium deoxycholate, 1 mM DTT, 1 mM phenylmethylsulfonyl fluoride (PMSF). Biotinylated proteins in the clarified lysates were purified using streptavidin magnetic beads (GE Healthcare Life Science), digested with trypsin (Promega), and desalted using SOLAμ SPE Plates (Thermo Fisher Scientific). Liquid chromatography with tandem mass spectrometry (LC-MS/MS) was performed using a Q Exactive HF-X mass spectrometer coupled to an Easy nLC 1200 system (Thermo Fisher Scientific). Peptides were separated using a 60-minute gradient on a homemade C18 column and ionized by electrospray. Full MS and HCD MS/MS spectra were acquired. Raw data were processed using MaxQuant

1.6.5.0, searching against the *S. japonicus* UniProt proteome (Proteome ID: UP000001744). Peptide and protein identifications were filtered to a 1% false discovery rate (FDR). Detailed parameters for LC-MS/MS and proteomics data have been deposited to the ProteomeXchange Consortium via the PRIDE partner repository with the dataset identifier PXD053432.

## Yeast two-hybrid assays

Yeast two-hybrid assays were performed using the pGBKT7 and pGADT7 plasmids that contain full-length coding sequences of the proteins of interest or those with specific mutations. The indicated plasmid pairs were co-transformed into the *S. cerevisiae* AH109 strain (*MATa, trp1–901, leu2–3, 112, ura3–52, his3–200, gal4Δ, gal80Δ, LYS2::GAL1*$_{UAS}$*-GAL1*$_{TATA}$*-HIS3, MEL1, GAL2*$_{UAS}$*-GAL2*$_{TATA}$*-ADE2, URA3:: MEL1*$_{UAS}$*-MEL1*$_{TATA}$*-lacZ*) (YC1010, Weidi Biotechnology Company). Successful co-transformation was confirmed by growth on complete minimal (CM) medium plates [108] lacking leucine and tryptophan (–Leu–Trp). Interactions were assessed by growth on CM medium lacking leucine, tryptophan, and histidine (–Leu–Trp–His) for medium interaction and on CM medium lacking leucine, tryptophan, histidine, and adenine (–Leu–Trp–His–Ade) for strong interactions.

A yeast reverse two-hybrid assay [78] was performed to identify critical amino acid residues of Ppn1 required for interaction with Clr4. Briefly, a cDNA fragment encoding Ppn1 was amplified by error-prone PCR using LA Taq DNA Polymerase (Takara Bio Inc.). The mutagenized *ppn1*$^+$ fragment, a gapped pGBKT7 plasmid, and pGADT7 carrying the *clr4*$^+$ gene were co-transformed into AH109. Transformants grown on CM–Leu–Trp plates were screened on CM–Leu–Trp–His plates for loss of growth, indicating loss of interaction. The *ppn1* mutations from clones exhibiting compromised Ppn1-Clr4 interaction were determined by Sanger sequencing.

## RNA immunoprecipitation (RNA-IP)

RNA-IP was conducted with anti-GFP (ab290, Abcam) antibody as described previously [39]. Specifically, yeast cells harvested during log-phase growth (OD$_{600}$ = 0.5–1) were washed with 1 × PBS containing 0.1 mM PMSF. Cells were lysed in RNA-IP buffer (50 mM HEPES, 140 mM NaCl, 10% glycerol, 1 mM EDTA, 0.1% Triton X-100, 0.1% NP- 40, 1 mM PMSF, 2 mM vanadyl ribonucleoside complex, 400 U/mL RNasin Plus RNase inhibitor) and 5% of the supernatant was saved as input after having been centrifuged. The remainder of the supernatant was then incubated with 40 µL of protein A/G magnetic beads (B23201, Bimake. com) coupled with 2 µL of anti-GFP antibody (ab290, Abcam) for 3 hours at 4°C with rotation. The beads were then separated and washed three times with RNA-IP buffer. RNA was cleaned up using the Trizol method, and DNA was removed through digestion with TURBO DNase (Thermo Fisher Scientific). Purified RNAs were again cleaned up using MasterPure Yeast RNA Purification Kit (Lucigen). RT-qPCR was performed using the PrimeScript II 1st strand cDNA Synthesis Kit (TaKaRa Bio Inc.) and ChamQ Universal SYBR qPCR Master Mix (Vazyme). Primer sequences used in RNA-IP are listed in S12 Table.

## Recombinant protein expression and purification

MBP-Pab2–6 × His tag and MBP-Pab2RA-6 × His tag fusion proteins were expressed and purified from *E. coli* BL21-CodonPlus (DE3)-RIL cells (EC1008, Weidi Biotechnology Company). *E. coli* cells were pre-cultured overnight at 37°C and inoculated at a ratio of 5–10% into 500ml of LB medium containing the appropriate antibiotics. *E. coli* cells were grown to OD$_{600}$ = 0.5–0.8 at 37°C, 150 rpm and induced with 1 mM IPTG at 37°C, 120 rpm for 2–4 hours. Pelleted cells were lysed in lysis buffer (50 mM pH 7.5 Tris–HCl, 1M NaCl, 1 M urea,

10 mM imidazole, 1.5 mM 2-mercaptoethanol, 1% NP-40, 5% glycerol and protease inhibitor cocktail) by sonication. Lysates were pelleted at 15,000 rpm at 4°C for 45 min. Supernatants were purified using 2 mL Ni NTA Beads (SA004025, Smart-Lifesciences) prewashed with lysis buffer at room temperature. Supernatants incubated with beads at 4°C for 1 hour and then beads were washed with lysis buffer for 6 times. Proteins were eluted with elution buffer (50 mM pH 7.5 Tris–HCl, 1M NaCl, 1 M urea, 500 mM imidazole, 1.5 mM 2-mercaptoethanol and 5% glycerol). The eluted proteins were analyzed by SDS-PAGE and dialyzed into dialysis buffer (20 mM pH 7.5 Tris–HCl, 150 mM NaCl, and 1 mM DTT). Proteins in dialysis buffer were concentrated by Amicon Ultra-4 Centrifugal Filter Unit (UFC803008, Sigma) and then incubated with TEV protease at 30°C for 1 hour. The fractions were analyzed by SDS-PAGE and stored at −80°C after flash frozen in liquid nitrogen.

## *In vitro* liquid droplet formation assay

After removing MBP and His tag in MBP-Pab2–6 × His tag and MBP-Pab2$^{RA}$-6 × His tag fusion proteins by TEV protease, proteins were diluted to the indicated concentrations with dialysis buffer (20 mM pH 7.5 Tris–HCl, 150 mM NaCl, and 1 mM DTT). Proteins were observed under a DIC microscope using a Zeiss Axio Imager Z2 microscope (Carl Zeiss MicroImaging) with a 40x objective. All imaged were captured within 5 min after LLPS induction at room temperature.

## Supporting information

**S1 Fig. Characterization of *S. japonicus pab2Δ*.** (A) Growth assay of wild-type (WT) and *pab2Δ* cells. Ten-fold serial dilutions were spotted onto complete medium plates and incubated at the indicated temperatures. (B) WT and *pab2Δ* cells were grown in complete liquid media, and OD$_{600}$ was measured at the indicated temperatures. Data represent mean ± SD of three independent experiments. (C) Microscopic analyses of DAPI-stained WT and *pab2Δ* cells. Scale bars, 2 μm. The percentage of cells with abnormal DNA and the number of cells counted are shown below each panel. (D) DIC images of WT and *pab2Δ* cells were taken after sporulation induction. Scale bars, 2 μm. (E) Homothallic WT and *pab2Δ* cells were subjected to a nitrogen-depleted condition, and mating efficiency was assessed by counting 2000 cells under a microscope. Scale bars, 2 μm. (F) Differentially expressed genes (DEGs) in *pab2Δ* (two biological replicates) are shown in Venn diagrams. The left diagram shows genes with > 2-fold increased expression, and the right diagram shows genes with < 0.5-fold decreased expression. For each comparison, the same set of 5,272 genes was examined. The statistical significance (*p*-value) of the overlap between each of the two groups using a hypergeometric test is shown under the diagrams. (G) Putative meiotic gene expression in WT and *red1Δ*. RT-qPCR analysis of four inferred meiotic genes in WT and *red1Δ* cells. The four putative meiotic mRNAs (*mei4*, *ssm4*, *rec8*, and *spo5*) were normalized to *act1* mRNA to determine their relative expression levels. Data are presented as mean ± SD from three independent experiments. Statistical significance was determined using a two-tailed unpaired *t*-test (*$p < 0.05$, **$p < 0.01$, and ***$p < 0.005$; N.S.: not significant). (H) The localization of Pab2-GFP and Red1-mCherry during vegetative growth was examined by fluorescent microscopy. Scale bars, 2 μm. (TIF)

**S2 Fig. Isolation and characterization of the *clr4–1* mutant strain.** (A) Schematic representation of the amplicons used in this study. The position of *cen* and *tel* amplicons, located within the pericentromeric heterochromatin of centromere 3 and the right end of chromosome 2 (Telomere 2R), respectively, is depicted. (B) Growth assay of wild-type (WT) and *clr4–1* cells. Ten-fold serial dilutions were spotted onto complete medium plates and incubated at

the indicated temperatures. (C) Schematic representation of the Clr4 protein and the mutations found in *clr4–1*. Clr4 has a chromodomain (Chromo) and a SET domain. The *clr4–1* allele contains two base substitutions (A153G and T1030C), and only the T1030C substitution resulted in the C334R substitution in the SET domain. (D) Venn diagrams show differentially expressed genes (DEGs) (left: > 2-fold increased expression; right: < 0.5-fold decreased expression) in *pab2Δ* and *clr4–1* cells. The statistical significance (*p*-value) of the overlap between each of the two groups using a hypergeometric test is shown under the diagrams. (E) GO analysis of the downregulated genes in *pab2Δ* and *clr4–1* cells. The Y-axis represents the GO annotations (e.g., gamma-aminobutyric acid metabolic process). The X-axis represents the significance of the enriched annotations (-$\log_{10}p$). (F) ChIP-qPCR analysis of RNA polymerase II (Pol II) occupancy at centromere 3 (*cen*) and telomere 2R (*tel*) in WT, *pab2Δ*, and *clr4–1*. Data are presented as mean ± SD from three independent experiments. Statistical significance was determined using a one-way ANOVA followed by Dunnett's multiple comparison test, with WT as the reference sample (*$p < 0.05$ and **$p < 0.01$; N.S.: not significant). (TIF)

**S3 Fig. Cid14 is involved in transcription sliencing and heterochromatin formation at heterochromatic regions.** (A) RT-qPCR analysis of centromeric and telomeric transcripts in WT, *cid14Δ*, *pab2*^RRM3A, and *cid14Δ-RRM3A* strains. Centromeric (*cen*) and telomeric (*tel*) transcripts were normalized to *act1* mRNA to determine their relative expression levels. Data are presented as mean ± SD from three independent experiments. Statistical significance was determined using a one-way ANOVA followed by Dunnett's multiple comparison test, with WT as the reference sample (*$p < 0.05$ and **$p < 0.01$; N.S.: not significant). (B) ChIP-qPCR analysis of H3K9me2 and total H3 enrichment in WT, *cid14Δ*, *pab2*^RRM3A, *cid14Δpab2*^RRM3A, and *pab2Δ*. Data are presented as mean ± SD from three independent experiments. Statistical significance was determined using a one-way ANOVA followed by Dunnett's multiple comparison test, with WT as the reference sample (*$p < 0.05$, **$p < 0.01$, ***$p < 0.001$, and ****$p < 0.0001$; N.S.: not significant). (TIF)

**S4 Fig. Characterization of Pab2 nuclear dots and the importance of the C-terminal intrinsically disordered domain.** (A) The number of Pab2 foci in vegetative cells expressing Pab2-GFP. A total of 200 cells were examined. (B) Pab2-GFP localization in untreated, treated with 5% (w/v) 1,6-hexanediol for 10 minutes (Treated) and after a 60-min recovery period following 1,6-hexanediol removal (Recovered). Scale bars, 2 μm. (C) Western blotting of Pab2-GFP before, during, and after 1,6-hexanediol treatment (untreated, treated, and recovered). Pcn1/PCNA was used as a loading control. (D) Schematic representation of Pab2 and the Pab2ΔIDR truncation mutant proteins. (E) Thiabendazole (TBZ) sensitivity assay of untagged (WT), *clr4–1*, Pab2ΔIDR-GFP, Pab2^RA-GFP, and Pab2-GFP cells. Ten-fold serial dilutions were spotted onto complete medium plates with or without TBZ and grown at 30°C. (F) RT-qPCR analysis of transcripts from *mei4*^+, *ssm4*^+, *spo5*^+, *rec8*^+, as well as centromeric (*cen*) and telomeric (*tel*) transcripts, in Pab2-GFP and Pab2ΔIDR-GFP strains. The transcripts were normalized to *act1* mRNA to determine their relative expression levels. Data are presented as mean ± SD from three independent experiments. Statistical significance was determined using a two-tailed unpaired *t*-test (*$p < 0.05$ and ***$p < 0.005$; N.S.: not significant). (G) Schematic representation of MBP-Pab2–6 × His tag and MBP-Pab2^RA-6 × His tag fusion proteins. (H) Purified MBP-Pab2–6 × His tag and MBP-Pab2^RA-6 × His tag fusion proteins with and without TEV treatment were stained by Coomassie brilliant blue. (I) RT-qPCR analysis of transcripts from *mei4*^+, *ssm4*^+, *spo5*^+, *rec8*^+, as well as centromeric (*cen*) and telomeric (*tel*) transcripts, in Pab2-GFP and Pab2^RA-GFP strains. The transcripts were normalized to *act1*

mRNA to determine their relative expression levels. Data are presented as mean ± SD from three independent experiments. Statistical significance was determined using a two-tailed unpaired *t*-test (**p* < 0.05 and ***p* < 0.01; N.S.: not significant). (J) TBZ sensitivity assay of WT, *clr4–1*, and *mit1Δ* cells. Ten-fold serial dilutions were spotted onto complete medium plates with or without thiabendazole and grown at 30°C.
(TIF)

**S5 Fig. Pab2 RNA recognition motif is involved in transcription silencing at heterochromatic regions.** (A) RT-qPCR analysis of transcripts from *mei4$^+$*, *ssm4$^+$*, *spo5$^+$*, *rec8$^+$*, as well as centromeric (*cen*) and telomeric (*tel*) transcripts, in WT, *pab2ΔRRM*, *pab2$^{RRM3A}$*, and *pab2$^{RRM3A-RA}$* strains. The transcripts were normalized to *act1* mRNA to determine their relative expression levels. Data are presented as mean ± SD from three independent experiments. (Left) Statistical significance was determined using a one-way ANOVA followed by Dunnett's multiple comparison test, with WT as the reference sample (**p* < 0.05, ***p* < 0.01, and ****p* < 0.001; N.S.: not significant) (Right) Statistical significance was determined using a two-tailed unpaired *t*-test (**p* < 0.05, ***p* < 0.01 and ****p* < 0.005; N.S.: not significant). (B) Thiabendazole (TBZ) sensitivity assay of untagged (WT), *clr4–1*, *pab2ΔRRM*, Pab2$^{RRM3A-RA}$-GFP, Pab2$^{RRM3A}$-GFP, and Pab2-GFP cells. Ten-fold serial dilutions were spotted onto complete medium plates with or without TBZ and grown at 30°C. (C) Fluorescent microscopy of Pab2-GFP and Chp1-tdTomato localization during vegetative growth. Scale bars, 2 μm.
(TIF)

**S6 Fig. Characterization of Pab2-binding proteins.** (A) Schematic representation of the Red5 protein and the mutations found in *red5–1*. Red5 has multiple CCCH-type zinc-finger motifs. The *red5–1* allele contains two amino acid substitutions, Q253R and T265A, in the zinc finger domain. (B) Growth assay of wild-type (WT) and *red5–1* cells. Ten-fold serial dilutions were spotted onto complete medium plates and incubated at 18°C and 30°C. (C) Thiabendazole (TBZ) sensitivity assay of the indicated strains. Ten-fold serial dilutions were spotted onto complete medium plates with or without TBZ and grown at 30°C. (D) RT-qPCR analysis of transcripts from *mei4$^+$*, *ssm4$^+$*, *spo5$^+$*, *rec8$^+$*, as well as centromeric (*cen*) and telomeric (*tel*) transcripts, in WT, *red5–1*, and *rmn1Δ* strains. The transcripts were normalized to *act1* mRNA to determine their relative expression levels. Data are presented as mean ± SD from three independent experiments. Statistical significance was determined using a one-way ANOVA followed by Dunnett's multiple comparison test, with WT as the reference sample (**p* < 0.05 and ***p* < 0.01; N.S.: not significant). (E) The Pab2 protein was separated into three domains, the N-terminal domain (N), the RNA recognition motif (RRM), and the C-terminal intrinsically disordered domain (IDR), and each fragment was subjected to yeast two-hybrid assays to test its interaction with Red5 and Rmn1. (F) Western blotting of Pab2-GFP and Pab2ΔN-GFP. Pcn1/PCNA was used as a loading control. (G) The localization of Pab2ΔN-GFP/Red5-tdTomato and Pab2ΔN-GFP/Rmn1-tdTomato was examined by fluorescent microscopy. Scale bars, 2 μm. (H) Growth and TBZ sensitivity assay of WT, *pab2Δ*, and *pab2ΔN* strains. Ten-fold serial dilutions were spotted onto complete medium plates with or without TBZ and incubated at 18°C and 30°C (for growth assay) or at 30°C (for TBZ sensitivity assay).
(TIF)

**S7 Fig. Dri1, an Rmn1-binding protein, is dispensable for constitutive heterochromatin formation.** (A) Yeast two-hybrid assays to examine the interaction between Dri1 and Pab2. Ten-fold serial dilutions of the *S. cerevisiae* AH109 strain carrying indicated plasmids were spotted onto minimal plates lacking leucine and tryptophan (-Leu-Trp), leucine, tryptophan, and histidine (–Leu–Trp–His), or leucine, tryptophan, histidine, and adenine

(–Leu–Trp–His–Ade). p53 and T antigen (T) were used as positive controls. (B) ChIP-qPCR analysis of H3K9me2 and total H3 enrichment at centromeres (*cen*) and telomeres (*tel*) using the indicated strains. Data are presented as mean ± SD from three independent experiments. Statistical significance was determined using a one-way ANOVA followed by Dunnett's multiple comparison test, with WT as the reference sample (**$p < 0.01$ and ***$p < 0.001$; N.S.: not significant). (C) Thiabendazole (TBZ) sensitivity assay of the indicated strains. Ten-fold serial dilutions were spotted onto complete medium plates with or without TBZ and incubated at 26°C.
(TIF)

**S8 Fig. Characterization of the Ppn1 mutant that does not interact with Clr4.** (A) Yeast two-hybrid assays testing the interaction between Ppn1 and the N-terminal (N), RNA recognition motif (RRM), intrinsically disordered domain (IDR), ΔN, ΔRRM, or ΔIDR of Pab2. Ten-fold serial dilutions of the *S. cerevisiae* AH109 strain carrying indicated plasmids were spotted onto minimal plates lacking leucine and tryptophan (–Leu–Trp), leucine, tryptophan, and histidine (–Leu–Trp–His), or leucine, tryptophan, histidine, and adenine (–Leu–Trp–His–Ade). p53 and T antigen (T) were used as positive controls. (B) The localization of Ppn1-tdTomato in vegetative WT and *pab2Δ* cells was examined by fluorescent microscopy. Scale bars, 2 μm. The percentage of cells which can form Ppn1 foci and the number of cells counted are shown below each panel. (C) Western blotting of Ppn1-tdTomato in WT and Pab2Δ strains. Pcn1/PCNA was used as a loading control. (D) Yeast two-hybrid assays for the assessment of the interaction between Ppn1 and three Clr4 fragments (CD: chromodomain, M: middle, and SET: the SET domain). The assays were performed as in (A). (E) Yeast two-hybrid assays for testing Ppn1-Clr4 and Ppn1$^{V354A, L355P}$-Clr4 interactions. The assays were conducted as in (A). (F) RT-qPCR analysis of transcripts from *mei4*$^+$, *ssm4*$^+$, *spo5*$^+$, *rec8*$^+$, as well as centromeric (*cen*) and telomeric (*tel*) transcripts, in WT and *ppn1*$^{V354A, L355P}$ strains. The transcripts were normalized to *act1* mRNA to determine their relative expression levels. Data are presented as mean ± SD from three independent experiments. Statistical significance was determined using a two-tailed unpaired *t*-test (*$p < 0.05$ and **$p < 0.01$; N.S.: not significant).
(TIF)

**S1 Table. Genes upregulated (>2) in *pab2Δ*.**
(XLSX)

**S2 Table. Genes downregulated (<0.5) in *pab2Δ*.**
(XLSX)

**S3 Table. GO analysis of genes with expression increased (>2) in *pab2Δ*.**
(XLSX)

**S4 Table. GO analysis of genes with decreased expression (<0.5) in *pab2Δ*.**
(XLSX)

**S5 Table. Genes upregulated (>2) in *clr4–1*.**
(XLSX)

**S6 Table. Genes downregulated (<0.5) in *clr4–1*.**
(XLSX)

**S7 Table. Genes upregulated (>2) in both *pab2Δ* and *clr4–1*.**
(XLSX)

**S8 Table. Genes downregulated (<0.5) in both *pab2Δ* and *clr4–1*.**
(XLSX)

**S9 Table. Mass spectrometry results (peptide counts and LFQ) for Pab2-TurboID, Rpn12-TurboID, and NIG2021 (an untagged control strain).**
(XLSX)

**S10 Table. Mutation sites in *pla1–3*.**
(XLSX)

**S11 Table. Strains used in this study.**
(XLSX)

**S12 Table. Primers used in this study.**
(XLSX)

**S13 Table. Underlying numerical data for all graphs.**
(XLSX)

**S1 Movie. Pab2 droplets *in vitro*.**
(AVI)

## Acknowledgment

We thank K. Aoki, E. Bayne, K. Furuya, F. Ishikawa, J. Liu, S. Oliferenko, R. Tsien, X. Tong, H. Tadakuma, M. Zhuang, and the National Institute of Genetics (Japan) for materials and technical advice, as well as the members of the T.S. lab for helpful discussions. We gratefully acknowledge the technical support provided by the Molecular Imaging Core Facility (MICF), Multi-Omics Core Facility (MOCF), and the Molecular and Cell Biology Core Facility (MCBCF) at the School of Life Science and Technology, ShanghaiTech University.

## Author contributions

**Data curation:** Gobi Thillainadesan.

**Formal analysis:** Ziyue Liu, Gobi Thillainadesan.

**Funding acquisition:** Tomoyasu Sugiyama.

**Investigation:** Ziyue Liu, Xiuyi Song.

**Supervision:** Tomoyasu Sugiyama.

**Visualization:** Ziyue Liu, Tomoyasu Sugiyama.

**Writing – original draft:** Tomoyasu Sugiyama.

**Writing – review & editing:** Ziyue Liu, Tomoyasu Sugiyama.

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
