## [Decision Letter · Decision Letter 0]

24 Feb 2025

PGENETICS-D-25-00088

The nuclear poly(A)-binding protein Pab2/PABPN1 promotes heterochromatin assembly through the formation of Pab2 nuclear condensates

PLOS Genetics

Dear Dr. Sugiyama,

Thank you for submitting your manuscript to PLOS Genetics. The manuscript has been reviewed by two of the original reviewers as well as a new reviewer (Reviewer 3). While the reviewers are generally supportive of publication in PLOS Genetics, Reviewer 3 has requested several minor revisions prior to acceptance, which we feel are critical meet PLOS Genetics's publication criteria. Therefore, we invite you to submit a revised version of the manuscript that addresses the points raised during the review process.

Please submit your revised manuscript within 30 days Mar 26 2025 11:59PM. If you will need more time than this to complete your revisions, please reply to this message or contact the journal office at plosgenetics@plos.org. Please include the following items when submitting your revised manuscript:

We look forward to receiving your revised manuscript.

Kind regards,

Sigurd Braun, PhD

Guest Editor

PLOS Genetics

Wendy Bickmore

Section Editor

PLOS Genetics

Aimée Dudley

Editor-in-Chief

PLOS Genetics

Anne Goriely

Editor-in-Chief

PLOS Genetics

**Additional Editor Comments :**

Please ensure that appropriate statistical tests for multiple comparisons are included in the experiments highlighted by Reviewer 3. Additionally, I recommend clearly indicating the reference group (i.e., the baseline for statistical comparisons), as this is particularly important for evaluating additive or synergistic effects (for example, in Figure S3B for cid14∆ and cid14∆-RRM3A), and making changes to the text if needed..

Furthermore, regarding Reviewer 3’s comments on the role of leo1, you may wish to revise this section by either incorporating relevant details from your previous manuscript or removing it entirely for clarity.

**Journal Requirements:**

1) We noticed that you used the phrase 'data not shown' in the manuscript. We do not allow these references, as the PLOS data access policy requires that all data be either published with the manuscript or made available in a publicly accessible database. Please amend the supplementary material to include the referenced data or remove the references. 2) We do not publish any copyright or trademark symbols that usually accompany proprietary names, eg ©,  ®, or TM  (e.g. next to drug or reagent names). Therefore please remove all instances of trademark/copyright symbols throughout the text, including:- ® on page: 49. 3) We have noticed that you have uploaded Supporting Information files, but you have not included a complete list of legends. Please add a full list of legends for your Supporting Information files (supplementary tables and movie) after the references list. 4) Thank you for stating that "Detailed parameters for LC-MS/MS and proteomics data have been deposited to the ProteomeXchange Consortium via the PRIDE partner repository with the dataset identifier PXD053432."  We strongly recommend all authors deposit their data before acceptance, as the process can be lengthy and hold up publication timelines. Please note that, though access restrictions are acceptable now, your entire minimal dataset will need to be made freely accessible if your manuscript is accepted for publication. This policy applies to all data except where public deposition would breach compliance with the protocol approved by your research ethics board. If you are unable to adhere to our open data policy, please kindly revise your statement to explain your reasoning and we will seek the editor's input on an exemption. 5) Please amend your detailed Financial Disclosure statement. This is published with the article. It must therefore be completed in full sentences and contain the exact wording you wish to be published.1) State the initials, alongside each funding source, of each author to receive each grant. For example: "This work was supported by the National Institutes of Health (####### to AM; ###### to CJ) and the National Science Foundation (###### to AM)."2) State what role the funders took in the study. If the funders had no role in your study, please state: "The funders had no role in study design, data collection and analysis, decision to publish, or preparation of the manuscript." 6) The following files are currently uploaded as file type 'Other', which are not viewable by the reviewers: "Fig_WB.tif" and "Fig_Y2H.tif ."  Please change the files type to 'Supporting Information' and include legends in the manuscript if you wish them to be included in review. 

**Reviewers' comments:**

Reviewer's Responses to Questions

Reviewer #1: The authors correctly answered my comments. In addition, their responses to the other reviewers’ comments improve the manuscript. I believe the manuscript is now ready for publication in Plos Genetics.

Reviewer #2: The new submission has addressed all my concerns and has improved significantly. I am supportive for its publication on PLoS Genetics

Reviewer #3: Heterochromatin formation has been extensively studied, revealing many insights, yet numerous uncertainties remain. In this manuscript by Liu et al., the authors demonstrate that PABPN1 plays a key role in the formation of constitutive heterochromatin in the fission yeast S. japonicus. Deletion of the pab2 gene reduces the centromeric localization of the H3K9 methyltransferase and the chromatin remodeler, and impairs centromeric heterochromatin formation. This study demonstrates that S. japonicus, along with S. pombe, can serve as an excellent model system to study the mechanisms of heterochromatin formation.

The authors have adequately addressed the reviewers' comments from the first round of review; however, there are still some points that should be further improved, and certain parts lack detailed description, as described below.

1_Has it been confirmed that the expression of genes such as mei4 and ssm4, which are referred to as meiotic genes, is truly meiosis-specific in S. japonicus? If not, please revise the wording in the text accordingly.

2_ While the ChIP-qPCR and RNA-IP data include information on normalization, there is no information provided on what the RT-qPCR data (Figure 1 and 2) were normalized to.

3_Please add a brief explanation to the text about tbp1 and cdc2 in Figure 2D.

4_Figure 3A. It would be helpful for general readers to have a more detailed explanation of the differences between H3K9me2 and me3.

5_Line506. The red5 knockout strain is not used; is it an essential gene?

6_Figure S7A, Lines 612-613. The figure does not clearly show an interaction between Rmn1 and Dri1.

7_Line846. There is no explanation of leo1 in the text, making it difficult to follow why leo1Δ is introduced.

8_ Multiple t-tests are being conducted in the qPCR analyses (Figure 2C, 3A, 3B, 3C 4K, 5D, 5E, 5F, 6D, 6E, 6F, 6H, 7B, 8H, S2F, S3, S5A, S6D, and S7B), which is inappropriate. ANOVA and multiple comparisons should be performed instead.

**Have all data underlying the figures and results presented in the manuscript been provided?**

Reviewer #1: Yes

Reviewer #2: Yes

Reviewer #3: Yes

PLOS authors have the option to publish the peer review history of their article (what does this mean? ). If published, this will include your full peer review and any attached files.

**Do you want your identity to be public for this peer review?** For information about this choice, including consent withdrawal, please see our Privacy Policy .

Reviewer #1: No

Reviewer #2: No

Reviewer #3: No

**Figure resubmission:**
---

## [Editor Report · Decision Letter 1]

8 Mar 2025

Dear Dr Sugiyama,

We are pleased to inform you that your manuscript entitled "The nuclear poly(A)-binding protein Pab2/PABPN1 promotes heterochromatin assembly through the formation of Pab2 nuclear condensates" has been editorially accepted for publication in PLOS Genetics. Congratulations!

Yours sincerely,

Sigurd Braun, PhD

Guest Editor

PLOS Genetics

Wendy Bickmore

Section Editor

PLOS Genetics

Aimée Dudley

Editor-in-Chief

PLOS Genetics

Anne Goriely

Editor-in-Chief

PLOS Genetics

Comments from the reviewers (if applicable):

**Data Deposition**

http://datadryad.org/submit?journalID=pgenetics&manu=PGENETICS-D-25-00088R1

**Press Queries**

---

## [Editor Report · Acceptance letter]

PGENETICS-D-25-00088R1

The nuclear poly(A)-binding protein Pab2/PABPN1 promotes heterochromatin assembly through the formation of Pab2 nuclear condensates

Dear Dr Sugiyama,

We are pleased to inform you that your manuscript entitled "The nuclear poly(A)-binding protein Pab2/PABPN1 promotes heterochromatin assembly through the formation of Pab2 nuclear condensates" has been formally accepted for publication in PLOS Genetics! Your manuscript is now with our production department and you will be notified of the publication date in due course.

With kind regards,

Zsofia Freund

PLOS Genetics

On behalf of:
